# WHERE WE HAVE ARRIVED IN PROVING THE EMERGENCE OF SPARSE INTERACTION PRIMITIVES IN DNNS

**Qihan Ren, Jiayang Gao, Wen Shen, Quanshi Zhang**[*]
Shanghai Jiao Tong University
{renqihan,gjy0515,adashen,zqs1022}@sjtu.edu.cn

## ABSTRACT

This study aims to prove the emergence of symbolic concepts (or more precisely, sparse primitive inference patterns) in well-trained deep neural networks (DNNs). Specifically, we prove the following three conditions for the emergence. (i) The high-order derivatives of the network output with respect to the input variables are all zero. (ii) The DNN can be used on occluded samples and when the input sample is less occluded, the DNN will yield higher confidence. (iii) The confidence of the DNN does not significantly degrade on occluded samples. These conditions are quite common, and we prove that under these conditions, the DNN will only encode a relatively small number of sparse interactions between input variables. Moreover, we can consider such interactions as symbolic primitive inference patterns encoded by a DNN, because we show that inference scores of the DNN on an exponentially large number of randomly masked samples can always be well mimicked by numerical effects of just a few interactions. The code is available at https://github.com/sjtu-xai-lab/interaction-sparsity.

## 1 INTRODUCTION

In the field of explainable AI, one of the fundamental problems is whether the inference logic of a deep neural network (DNN) can really be explained as symbolic concepts. Although some interesting phenomena of the emergence of concepts in DNNs have been discovered in previous studies (Li & Zhang, 2023b; Ren et al., 2023a), *the core problem is still not strictly formulated or proven, i.e., strictly proving whether the knowledge encoded by DNNs is indeed symbolic.*

To this end, it is a significant challenge to prove whether or not the knowledge encoded by a black-box DNN is symbolic. Up to now, heuristic studies usually explained DNN features using manually annotated concepts (Bau et al., 2017; Kim et al., 2018), without formally defining or proving the concepts in a DNN. Thus, the proof of the emergence of symbolic concepts in DNNs will have a profound impact on both theory and practice.

However, the definition of concepts encoded by a DNN is still an open problem, because it is a complex cross-disciplinary issue related to cognitive science, neuroscience, and mathematics. Nevertheless, let us ignore cognitive issues and limit our discussion to the scope of *whether we can obtain a relatively small set of primitive inference patterns to strictly explain the complex changes of inference scores of the DNN on different input samples.*

To be precise, we hope to prove a set of sufficient conditions to enable us to represent the knowledge of a DNN as symbolic primitives. To this end, if we ignore cognitive issues, Ren et al. (2023a) and Li & Zhang (2023b) have considered Harsanyi interactions (Harsanyi, 1963) to represent the symbolic primitives encoded by a DNN. It is because they have empirically observed that a well-trained DNN usually only encoded a few salient interactions, and they have also observed that the network output could be well mimicked by these interactions. Specifically, each interaction represents an AND relationship between a set $S$ of input variables, and this interaction has a numerical effect $I(S)$ on the inference score of the DNN. The input variables can be image regions for image classification or words for natural language processing. As Figure 1 shows, when classifying a dog

---

[*]Quanshi Zhang is the corresponding author. He is with the Department of Computer Science and Engineering, the John Hopcroft Center, at the Shanghai Jiao Tong University, China.

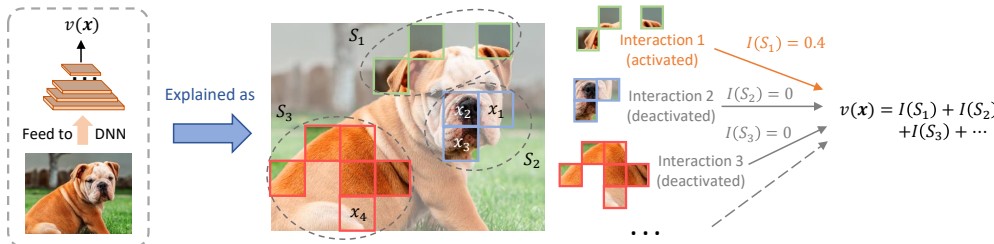

Figure 1: Illustration of interactions encoded by a DNN. Each interaction $S$ corresponds to an AND relationship among a specific set $S$ of input variables (image patches). The patches $x_1$ and $x_4$ are masked, so that interactions $S_2$ and $S_3$ are deactivated.

image, a DNN may encode a salient interaction among three image regions in $S = \{x_1, x_2, x_3\}$. Only when all three image regions are present will the interaction be activated and contribute a numerical effect $I(S)$ to the inference score of the DNN. Masking any regions in $S$ will deactivate the interactions and make $I(S) = 0$.

Therefore, the proof of the emergence of interaction primitives is to prove a set of common conditions, which makes a DNN only encode **a small number** of symbolic (sparse) interactions. It means that given an input sample $x$, the inference score $v(x)$ of the DNN can be disentangled into the sum of the effects $I(S)$ of a few salient interactions, *i.e.*, $v(x) \approx \sum_{S \in \Omega_{\text{salient}}} I(S) + bias$, which can be called **interaction primitives**. Specifically, we prove three conditions. (i) The DNN has at most $M$-th order non-zero derivatives, where $M < n$, and $n$ is the number of input variables to the DNN. (ii) The DNN can be used on occluded samples (*e.g.*, an image with some patches being masked), and yields a higher classification confidence when the sample is less occluded. (iii) The classification confidence of the DNN does not significantly degrade on occluded samples. Because these conditions are common for many DNNs, and the proof does not depend on the specific architecture of the network, our proof ensures that the emergence of symbolic interaction primitives is a *universal phenomenon* for various networks trained for different tasks.

In fact, the essential reason for the emergence of sparse interaction primitives is **neither** the architecture of the neural network, **nor** the sparsity of network parameters/intermediate-layer features, but the property of the task. To be precise, when the task requires the DNN to conduct smooth inference on masked/occluded samples, a well-trained DNN usually encodes sparse interaction primitives.

## 2 RELATED WORK

We have developed a system of game-theoretic interactions for explaining DNNs in the last three years, and have published more than fifteen papers in this direction. This system focused on addressing the following problems in explainable AI: (i) explicitly defining, extracting, and counting interactions encoded by a DNN, (ii) explaining the representation capacity (*e.g.*, generalization ability and adversarial robustness) of DNNs from the perspective of interaction, and (iii) summarizing/explaining common mechanisms shared by different empirical deep learning methods.

● *Explicitly defining and extracting interactions encoded by a DNN.* A representative approach in explainable AI was to explain the interactions between different input variables (Sundararajan et al., 2020; Tsai et al., 2022). Based on game theory, Zhang et al. (2021a;b; 2020) defined multi-variate interaction and multi-order interaction. Ren et al. (2023a) discovered the sparsity of interactions in experiments. Li & Zhang (2023b) further discovered that salient interactions were usually transferable across different input samples and exhibited certain discrimination power. Chen et al. (2024) extracted generalizable interactions shared by different DNNs. These studies indicated that salient interactions could be considered as interaction primitives encoded by a DNN. Furthermore, Ren et al. (2023b) used the sparsity of interactions to define the optimal baseline value for the Shapley value. Cheng et al. (2021a) used interactions of different complexities to explain the encoding of specific types of shapes and textures in DNNs for image classification. Cheng et al. (2021b) discovered that salient interactions usually represented prototypical visual patterns in images.

● *Explaining the representation capacity of DNNs using game-theoretic interactions.* Game-theoretic interactions have been used to explain the representation capacity of a DNN, although the following studies used the multi-order interaction (Zhang et al., 2020), rather than the Harsanyi

interaction. Nevertheless, the Harsanyi interaction has been proven to be compositional elements in the multi-order interaction. The multi-order interaction has been used to explain the adversarial robustness (Ren et al., 2021; Zhou et al., 2024), adversarial transferability (Wang et al., 2021), and generalization ability (Zhang et al., 2020; Zhou et al., 2024) of a DNN. In addition, Deng et al. (2021) proved the difficulty of a DNN in encoding middle-complexity interactions. Ren et al. (2023c) proved that compared to a standard DNN, a Bayesian neural network (BNN) tended to avoid encoding complex Harsanyi interactions. Liu et al. (2023) explained the intuition that DNNs learned simple Harsanyi interactions more easily than complex Harsanyi interactions.

• *Summarizing common mechanisms for the success of various empirical deep learning methods.*
Deng et al. (2024) found that the computation of attribution values for fourteen attribution methods could all be explained as a re-allocation of interaction effects. Zhang et al. (2022a) proved that twelve methods to improve the adversarial transferability in previous studies essentially shared the common utility of reducing the interactions between adversarial perturbation units.

## 3 DNNs TEND TO ENCODE SPARSE INTERACTIONS

### 3.1 OVERVIEW OF THE EMERGENCE OF SYMBOLIC (SPARSE) INTERACTIONS IN DNNs

In this study, we aim to prove that symbolic (sparse) interactions usually emerge in a well-trained DNN. Recent studies (Ren et al., 2023a; Li & Zhang, 2023b) have empirically discovered the emergence of symbolic (sparse) interactions in various DNNs on different tasks. In mathematics, the emergence of symbolic (sparse) interactions means that the DNN's inference logic can be represented as the detection of a small number of interactions with a certifiably low approximation error.

• **Definition.** A clear definition of interactions encoded by a DNN is required. In this study, the interaction is defined as the Harsanyi dividend (Harsanyi, 1963). Let us consider a trained DNN $v$ and an input sample $\boldsymbol{x} = [x_1, \cdots, x_n]^\top$ with $n$ input variables indexed by $N = \{1, \cdots, n\}$. The input variables can be image regions for image classification or words in an input sentence for a natural language processing task. Furthermore, without loss of generality, let us assume that the output of the DNN on the sample $\boldsymbol{x}$ is a scalar, denoted by $v(\boldsymbol{x}) \in \mathbb{R}$. For DNNs with multi-dimensional output, we may choose one dimension of the output vector as the final output $v(\boldsymbol{x})$. In particular, for classification tasks, we set $v(\boldsymbol{x}) = \log \frac{p(y=y^{\text{truth}}|\boldsymbol{x})}{1-p(y=y^{\text{truth}}|\boldsymbol{x})}$ by following Deng et al. (2021).

The interaction effect of the Harsanyi dividend (Harsanyi, 1963) (or the *Harsanyi interaction*) between a set $S \subseteq N$ of input variables is defined as follows:

$$I(S) \stackrel{\text{def}}{=} \sum\nolimits_{T \subseteq S} (-1)^{|S|-|T|} \cdot u(T), \tag{1}$$

where $u(T) \stackrel{\text{def}}{=} v(\boldsymbol{x}_T) - v(\boldsymbol{x}_\emptyset)$. Here, $v(\boldsymbol{x}_T)$ denotes the network output on the masked input sample $\boldsymbol{x}_T$, where the variables in $N \setminus T$ are masked using their baseline values $\boldsymbol{b} = [b_1, \ldots, b_n]^\top$, and the variables in $T$ are unchanged. Accordingly, $v(\boldsymbol{x}_\emptyset)$ denotes the network output on the sample $\boldsymbol{x}_\emptyset$, where all the input variables are masked. In particular, we have $u(\emptyset) = I(\emptyset) = 0$.

Each interaction can be understood as an *AND relationship* between input variables in $S$. For example, as Figure 1 shows, to recognize a dog face, let a DNN encode the collaboration between three image regions in $S = \{x_1, x_2, x_3\}$. Only when the image regions $x_1$, $x_2$, and $x_3$ are all present will the interaction be activated and make a certain numerical effect $I(S)$ on the network output. The absence (masking) of any of the three image regions (*e.g.*, the masking of $x_1$ in Figure 1) deactivates the interaction and removes the numerical effect, *i.e.*, $I(S) = 0$.

We followed Li & Zhang (2023b) to visualize interactions encoded by PointNet++ (Qi et al., 2017b) on the ShapeNet (Yi et al., 2016) dataset. Each point cloud in this dataset contained 2500 points. To simplify the visualization, Li & Zhang (2023b) considered each semantic part[1] on the point cloud as a *single* input variable to the DNN. Figure 2 shows that the interaction $S = \{light, front\ wheel\}$ usually contributed a positive effect to the network output (*i.e.*, $I(S) > 0$), whereas the interaction $S = \{front\ wheel, back\ frame, mid\ frame\}$ usually had a negative effect (*i.e.*, $I(S) < 0$).

• **Faithful compositional inference patterns.** We use the following three axiomatic properties to define interactions as representations of faithful inference patterns (or concepts) encoded by a DNN.

---

[1]Annotations of the semantic parts were provided by the ShapeNet dataset. See Appendix C.2 for details.

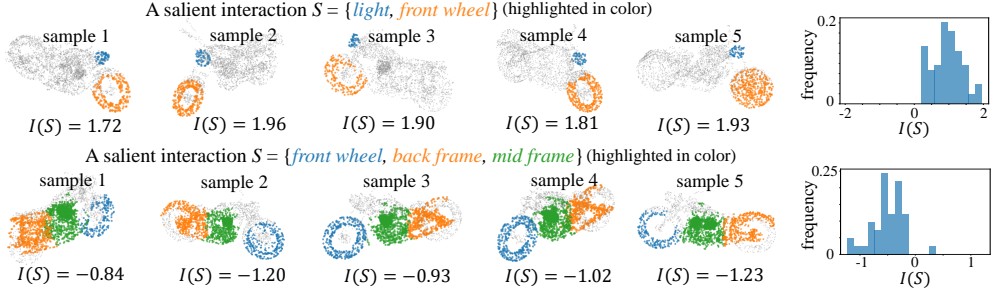

Figure 2: Interactions extracted by PointNet++ on different samples in the ShapeNet dataset and their corresponding effects $I(S)$. Histograms on the right show the distribution of interaction effects $I(S)$ on different "motorbike" samples.

(1) *Sparsity property.* A DNN is supposed to encode few salient interactions on a specific sample.

(2) *Universal matching property.* The network output on any arbitrarily masked sample is supposed to be well matched by the effects of specific interactions.

(3) *Sample-wise transferability property.* Salient interactions are supposed to be shared across different samples in the same category.

Li & Zhang (2023b) have empirically discovered the *sparsity* of Harsanyi interactions. They have also observed a significant overlap between salient interactions extracted from different samples in the same category, which indicated the *transferability* of interactions across different samples. In addition, it has been proven that the Harsanyi interaction satisfies the *universal matching* property.

**Theorem 1** (Proven in Ren et al. (2023a) and Appendix B.1). *Let the input sample $\boldsymbol{x}$ be arbitrarily masked to obtain a masked sample $\boldsymbol{x}_S$. The output of the DNN on masked sample $\boldsymbol{x}_S$ can be disentangled into the sum of all interaction effects within $S$: $\forall S \subseteq N$, $v(\boldsymbol{x}_S) = \sum_{T \subseteq S} I(T) + v(\boldsymbol{x}_\emptyset)$.*

Furthermore, the Harsanyi interaction has been proven to satisfy seven desirable properties in game theory, *e.g.*, the *efficiency, linearity, dummy*, and *symmetry* properties. In addition, the Harsanyi interaction can explain the elementary mechanism of three existing game-theoretic attribution/interaction metrics, *e.g.*, the Shapley value [2] (Shapley, 1953). See Appendix A for details.

● **Illustrating the emergence of interaction primitives.** The core task of proving the emergence of symbolic primitives is to prove that in a well-trained DNN, the interactions defined above are sparse. Although there are as many as $2^n$ interactions corresponding to all subsets $S$ in $2^N = \{S : S \subseteq N\}$, recent studies (Ren et al., 2023a; Li & Zhang, 2023b) have empirically discovered that these interactions were usually sparse in a well-trained DNN. That is, only a few salient interactions have significant effects on the network output and can be taken as **salient interaction primitives**. In comparison, most other interactions have near-zero effects (*i.e.*, $I(S) \approx 0$), which are referred to as **noisy patterns**. According to Theorem 1, the network output can be summarized by a small number of salient interactions, *i.e.*, $v(\boldsymbol{x}) \approx \sum_{S \in \Omega_{\text{salient}}, S \neq \emptyset} I(S) + v(\boldsymbol{x}_\emptyset)$.

Although the sparsity of interactions has already been demonstrated in previous studies (Ren et al., 2023a; Li & Zhang, 2023b), we still conducted experiments in this study to better illustrate this phenomenon. We followed Li & Zhang (2023b) to conduct experiments on various neural networks[3], including multilayer perceptrons (MLPs), residual MLPs (ResMLPs) (Touvron et al., 2022), LeNet (LeCun et al., 1998), AlexNet (Krizhevsky et al., 2012), VGG (Simonyan & Zisserman, 2014), ResNet (He et al., 2016), PointNet (Qi et al., 2017a), PointNet++ (Qi et al., 2017b), and on different datasets, including tabular data (the *tic-tac-toe* dataset (Dua & Graff, 2017), *phishing* dataset (Dua & Graff, 2017), and *wifi* dataset (Dua & Graff, 2017)), point cloud data (the *ShapeNet* dataset (Yi et al., 2016)), and image data (the *MNIST-3* dataset (LeCun et al., 1998) and *CelebA-eyeglasses* dataset (Liu et al., 2015)). For better visualization, we drew a curve of the interaction strength by normalizing the interaction $\tilde{I}(S) = I(S)/\max_{S'} |I(S')|$ in descending order. Figure 3 shows the strength curve that was averaged over different samples in the dataset, which was

---

[2]We have $\phi(i) = \sum_{S \subseteq N \setminus \{i\}} \frac{1}{|S|+1} I(S \cup \{i\})$, which means that the Shapley value $\phi(i)$ can be explained as the result of uniformly assigning each Harsanyi interaction $I(S \cup \{i\})$ to each involved variable.

[3]Please see Appendix C.1 for details on these DNNs and datasets.

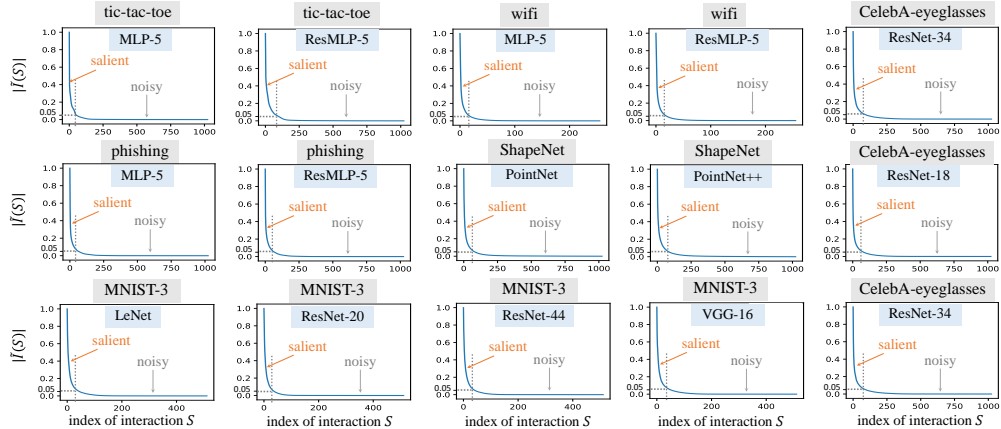

Figure 3: Normalized strength of different interactions, shown in descending order. Various DNNs trained for different tasks all encoded sparse interactions. In other words, only a relatively small number of interactions had a significant effect, while most interactions were noisy patterns and had near-zero effects, *i.e.*, $I(S) \approx 0$.

computed according to the method in (Li & Zhang, 2023b). Figure 3 successfully verifies that interactions encoded by various DNNs on different tasks were all sparse. Moreover, we followed Li & Zhang (2023b) to define interactions with $|I(S)| > \tau = 0.05 \cdot \max_{S'} |I(S')|$ as salient interactions.

## 3.2 PROVING THE SPARSITY OF INTERACTIONS

Despite these achievements, there is still no theory to prove the emergence of such sparse interactions as symbolic primitives. Therefore, in this subsection, we make an initial attempt to theoretically prove the sparsity of interactions. To be precise, **we need to prove the conditions, under which the output of a DNN can be approximated by the effects of *a small number* of salient interactions, instead of a mass of fuzzy features.** More interestingly, the proof will show that the sparsity of interactions depends on the property of the classification task itself, rather than the architecture of the neural network, or the sparsity of the parameters/intermediate-layer features.

Given a trained DNN $v$ and an input sample $\boldsymbol{x} = [x_1, \cdots, x_n]^\top$, let $v(\boldsymbol{x}) \in \mathbb{R}$ denote the scalar network output on the sample $\boldsymbol{x}$. Let us consider the Taylor expansion of the network output $v(\boldsymbol{x})$, which is expanded at the point $\boldsymbol{b} = [b_1, \cdots, b_n]^\top$:

$$v(\boldsymbol{x}) = \sum_{\kappa_1=0}^{\infty} \cdots \sum_{\kappa_n=0}^{\infty} \frac{\partial^{\kappa_1+\cdots+\kappa_n} v}{\partial x_1^{\kappa_1} \cdots \partial x_n^{\kappa_n}}\bigg|_{\boldsymbol{x}=\boldsymbol{b}} \cdot \frac{(x_1-b_1)^{\kappa_1} \cdots (x_n-b_n)^{\kappa_n}}{\kappa_1! \cdots \kappa_n!}. \tag{2}$$

Strictly speaking, there are no high-order derivatives for ReLU neural networks. In this case, we can still use the finite difference method (Peebles et al., 2020; Gonnet & Scholl, 2009) to compute the equivalent high-order derivatives yielded by the change in ReLU gating states.

**Assumption 1-$\boldsymbol{\alpha}$.** *Interactions higher than the $M$-th order have zero effect, i.e., $\forall S \in \{S \subseteq N \mid |S| \geq M+1\}$, $I(S) = 0$. The order is defined as the number of input variables in $S$,* $\text{order}(S) \stackrel{\text{def}}{=} |S|$.

**Assumption 1-$\boldsymbol{\beta}$.** *The network is assumed to have at most $M$-order non-zero derivatives, i.e., $\forall \boldsymbol{b} \in \mathbb{R}^n$, $\forall \kappa_1 \cdots \kappa_n \in \mathbb{N}$, s.t. $\kappa_1 + \cdots + \kappa_n \geq M+1$, we have $\frac{\partial^{\kappa_1+\cdots+\kappa_n} v}{\partial x_1^{\kappa_1} \cdots \partial x_n^{\kappa_n}}\big|_{\boldsymbol{x}=\boldsymbol{b}} = 0$.*

In mathematics, Assumption 1-$\alpha$ can be derived (see Appendix B.2) from Assumption 1-$\beta$ (assuming no derivatives higher than the $M$-th order in the Taylor expansion of the function $v$), which is a stronger assumption than Assumption 1-$\alpha$ and is specific to continuously differentiable functions[4]. In comparison, Assumption 1-$\alpha$ can be applied to more general networks than Assumption 1-$\beta$.

In fact, only Assumption 1-$\alpha$ without Assumption 1-$\beta$ is already enough to conduct our later proof. The assumed non-existence of high-order interactions in Assumption 1-$\alpha$ is not strange, and it has been illustrated in many large language models (please see Figure 4). This is because high-order

---

[4]This is a common setting for a simple DNN or a DNN with almost zero high-order derivatives. In fact, it is not necessary for most DNNs to have derivatives of very high orders during training.

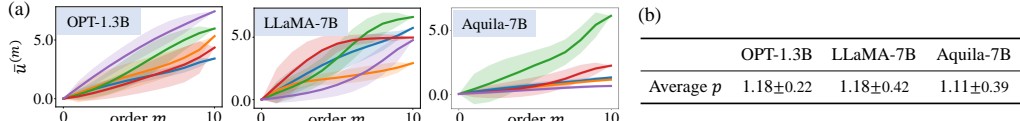

Figure 4: Box-and-whisker diagram for the strength of interactions $|I(S)|$ of each order $m$. We tested different LLMs (OPT-1.3B, LLaMA-7B, and Aquila-7B) on the SQuAD dataset. Experiments show that high-order interactions on these networks were usually close to zero. Please see Appendix D.1 for experimental details and Appendix E.1 for results on more samples.

Figure 5: (a) Visualization of the monotonicity. Each curve shows the monotonic increase of the average output of the $m$-th order $\bar{u}^{(m)}$ with the order $m$ on a sample. The shaded area indicates the standard deviation of all $m$-order outputs on a sample, *i.e.*, $\mathrm{Std}_{|S|=m}[u(S)]$. Note that the value of standard deviation does not affect our proof, because the proof only relies on the average output $\bar{u}^{(m)}$. (b) The average value of $p$ over different input samples, along with the standard deviation.

interactions usually represent extremely complex patterns and are usually unnecessary in real applications. For example, let us consider an interaction with a nonzero effect $I(S)$ corresponding to the AND relationship between $M' = 100 > M$ input variables. This indicates that if any of the 100 input variables is masked, then the interaction will be deactivated. Such an elaborate interaction is usually considered as an over-fitted pattern in real applications and it is not commonly learned by a DNN. In addition, ignoring very high-order interactions is common in the literature on game-theoretic interactions (Sundararajan et al., 2020; Tsai et al., 2022). In spite of that, we admit that there are a few special cases where high-order interactions may appear in real applications, but in those cases, extensive high-order interactions can be summarized as a simple effect, thereby not hurting the proof of the sparsity of interactions. Please see Section 3.3 for further discussion.

Second, let us consider a classification task in a real-world application, where some input samples may be partially occluded or masked. In fact, the classification of occluded samples is quite common, and a well-trained DNN is supposed to yield higher classification confidence for samples that are less masked. Therefore, we make the following monotonicity assumption:

**Assumption 2** (Monotonicity). *The average network output is assumed to monotonically increase with the size of the unmasked set $S$ of the input variables, i.e., $\forall\, m' \leq m$, we have $\bar{u}^{(m')} \leq \bar{u}^{(m)}$, where $\bar{u}^{(m)} \overset{\text{def}}{=} \mathbb{E}_{|S|=m}[u(S)]$, $u(S) = v(\boldsymbol{x}_S) - v(\boldsymbol{x}_\emptyset)$.*

This assumption indicates that the average classification confidence of the DNN increases when the input sample is less occluded. Specifically, $\bar{u}^{(m)} = \mathbb{E}_{|S|=m}[v(\boldsymbol{x}_S) - v(\boldsymbol{x}_\emptyset)] \geq 0$ represents the average network output over all masked samples $\boldsymbol{x}_S$ with $|S| = m$. We refer to $\bar{u}^{(m)}$ as *the average output of the $m$-th order* in the following discussion.

**Justification of the monotonicity assumption.** Assumption 2 commonly holds on different samples and DNNs. We justify this assumption through both theoretical analysis and experiments. Theoretically speaking, if we limit our discussion to two specific masked samples $S$ and $S'$ *s.t.* $S \subsetneq S'$, we cannot always ensure $u(S) < u(S')$, because some variables in $S' \setminus S$ may be unrelated to the classification, and may even have negative contributions to the classification. However, Assumption 2 focuses on the **average** output over all $m$-order masked states, $\bar{u}^{(m)} = \mathbb{E}_{|S|=m}[u(S)]$, which is much stabler and more robust than the output on a specific masked state $u(S)$. Thus, the monotonicity assumption is common for an input sample. Please see Appendix H for an example.

For experimental justification of Assumption 2, we conducted experiments on three Large Language Models (LLMs), including the OPT-1.3B model (Zhang et al., 2022b), the BAAI Aquila-7B model (FlagAI, 2023), and the LLaMA-7B model (Touvron et al., 2023). We tested these networks on 1000 sentences from the SQuAD dataset (Rajpurkar et al., 2016). For each input sentence, we followed Shen et al. (2023) to select $n = 10$ words from the sentence as input variables to reduce the computational cost. Please see Appendix D.1 for details on the input sentences, how to determine

Table 1: Our theory can explain the sparsity of interactions in about 84.52% - 89.87% samples, because these samples had monotonic values of $\bar{u}^{(m)}$. Nevertheless, the remaining 10.13% -15.48% samples without the monotonicity also encoded very sparse interactions. Our proof is still significant enough, although we cannot prove the sparsity on all input samples.

|  |  | OPT-1.3B | LLaMA-7B | Aquila-7B |
|---|---|---|---|---|
| Percent of samples with monotonicity |  | 89.87% | 84.52% | 87.46% |
| Avg # of valid interactions | Samples with monotonicity | 29.06±52.46 | 50.71±40.35 | 30.16±26.22 |
|  | Samples without monotonicity | 42.67±38.87 | 79.71±59.43 | 65.94±69.99 |

the set $N$, and how to mask words in $N \setminus T$ when computing $v(\boldsymbol{x}_T)$. Table 1 shows that on all LLMs, over 84% of the input samples satisfied the monotonicity assumption. Figure 5(a) also visualizes the increase of the average output of the $m$-th order $\bar{u}^{(m)}$ along with the order $m$ on different LLMs.

Table 1 also shows that because of the complexity of training an LLM, in a few cases, the average network output of the $m$-th order was not always monotonic along with the order $m$. In fact, this conflicted with the common logic, but it was still possible because the extremely high diversity of sentences made the LLM difficult to be sufficiently trained (fully converge). In this study, we did not consider such a case and only focused on the ideal case where a network was well-trained.

Third, we need to ensure that the classification confidence of the DNN does not significantly degrade on masked input samples. In real applications, the classification/detection of occluded samples is common. Thus, for a well-trained DNN, its confidence when classifying occluded (masked) samples should not be substantially lower than its confidence for unmasked samples.

**Assumption 3.** Given the average network output of samples with $m$ unmasked input variables, $\bar{u}^{(m)}$, we assume a lower bound for the average network output of samples with $m'$ ($m' \leq m$) unmasked input variables, *i.e.*, $\forall m' \leq m$, $\bar{u}^{(m')} \geq (\frac{m'}{m})^p \bar{u}^{(m)}$, where $p > 0$ is a positive constant.

In Assumption 3, we bound the decrease of the average output of order $m$ by a polynomial of degree $p$. If this assumption is violated, *i.e.*, $\bar{u}^{(m')} < (\frac{m'}{m})^p \bar{u}^{(m)}$ for $m' \leq m$, then it implies either extremely low classification confidence on masked samples or extremely high classification confidence on normal (unmasked) samples, which are both undesirable cases in real applications.

We also conducted experiments to illustrate the value of $p$ on real DNNs and datasets. We used the same LLMs and dataset as those in the previous experiment. Figure 5(b) shows that the value of $p$ across different samples was around 0.9 to 1.5, which was reasonable for further analysis.

**Proof of the sparsity of interactions.** Under the above conditions, we prove that interactions encoded by a DNN are sparse. We start by analyzing the upper bound of the sum of the effects of all $k$-order interactions, denoted by $A^{(k)} \stackrel{\text{def}}{=} \sum_{S \subseteq N, |S|=k} I(S)$.

**Theorem 2** (Proven in Appendix B.3). *There exists $m_0 \in \{n, n-1, \cdots, n-M\}$, such that for all $1 \leq k \leq M$, the sum of the effects of all $k$-order interactions can be written as*

$$A^{(k)} = (\lambda^{(k)} n^{p+\delta} + a^{(k)}_{\lfloor p \rfloor - 1} n^{\lfloor p \rfloor - 1} + \cdots + a^{(k)}_1 n + a^{(k)}_0) \, \bar{u}^{(1)}, \tag{3}$$

*where $|\lambda^{(k)}| \leq 1$, $|a^{(k)}_0| < n$, $|a^{(k)}_i| \in \{0, 1, \cdots, n-1\}$ for $i = 1, \cdots, \lfloor p \rfloor - 1$, and*

$$\delta \leq \log_n \left( \frac{1}{\lambda} \left( 1 - \frac{a_{\lfloor p \rfloor - 1}}{n^{p-\lfloor p \rfloor + 1}} - \cdots - \frac{a_0}{n^p} \right) \right), \quad \text{if } \lambda > 0, \tag{4}$$

$$\delta \leq \log_n \left( \frac{1}{-\lambda} \left( \frac{a_{\lfloor p \rfloor - 1}}{n^{p-\lfloor p \rfloor + 1}} + \cdots + \frac{a_0}{n^p} \right) \right), \quad \text{if } \lambda < 0. \tag{5}$$

*Here, $\lambda \stackrel{\text{def}}{=} \sum_{k=1}^{M} \frac{\binom{m_0}{k}}{\binom{n}{k}} \lambda^{(k)} \neq 0$, $a_i \stackrel{\text{def}}{=} \sum_{k=1}^{M} \frac{\binom{m_0}{k}}{\binom{n}{k}} a^{(k)}_i$ for $i = 0, 1, \cdots, \lfloor p \rfloor - 1$, and $\lfloor p \rfloor$ denotes the greatest integer that is less than or equal to $p$.*

The above theorem indicates that the sum of effects of all $k$-order interactions is $\mathcal{O}(n^{p+\delta})$. Then, we discuss the sparsity of interactions in the following two cases.

● **Case 1: When the positive and negative interactions of the $k$-th order do not fully cancel out each other.** Let $\eta^{(k)} \stackrel{\text{def}}{=} \frac{\sum_{|S|=k} I(S)}{\sum_{|S|=k} |I(S)|}$ denote the remaining proportion of the effects of $k$-order

Table 2: Comparison between the number of valid interactions and the derived upper bound.

|  | OPT-1.3B | LLaMA-7B | Aquila-7B | MLP (tabular dataset) |
|---|---|---|---|---|
| Real # of valid interactions | $28.73\pm_{52.37}$ | $50.53\pm_{40.37}$ | $30.13\pm_{26.20}$ | $54.42\pm_{36.81}$ |
| Upper bound | $197.84\pm_{188.87}$ | $293.20\pm_{287.28}$ | $184.23\pm_{124.71}$ | $229.11\pm_{139.52}$ |

interactions that are not cancelled out. Here, the absolute value of this proportion $|\eta^{(k)}|$ should not be negligible. Thus, let us set $|\eta^{(k)}| \gg \frac{1}{n}$. Otherwise, we consider that the positive and negative interactions *almost cancel out*, and then this instance belongs to Case 2.

*Without loss of generality, we set a small positive threshold $\tau$ subject to $0 < \tau \ll \mathbb{E}_{T \subseteq S}[|u(T)|]$. Then, we consider all interactions with $|I(S)| \geq \tau$ as valid (salient) interactions. We consider all interactions with $|I(S)| < \tau$ as noisy patterns.* In fact, as Figure 3 shows, most interactions below the threshold $\tau$ had roughly exponentially decreasing strength, which meant that most non-salient interactions (noisy patterns) had almost zero effect.

Therefore, let us use $R^{(k)} \overset{\text{def}}{=} |\{S \subseteq N \mid |S| = k, |I(S)| \geq \tau\}|$ to denote the number of valid interactions of the $k$-th order. The following theorem provides an upper bound for $R^{(k)}$.

**Theorem 3** (Proven in Appendix B.4). *$R^{(k)}$ has the following upper bound:*

$$R^{(k)} \leq \frac{\bar{u}^{(1)}}{\tau|\eta^{(k)}|}|\lambda^{(k)}n^{p+\delta} + a_{\lfloor p \rfloor -1}^{(k)}n^{\lfloor p \rfloor -1} + \cdots + a_0^{(k)}|. \tag{6}$$

The above theorem indicates that if positive interactions do not fully cancel with negative interactions (*i.e.*, $|\eta^{(k)}|$ is not extremely small), then the number of valid interactions $R^{(k)}$ of the $k$-th order has an upper bound of $\mathcal{O}(n^{p+\delta}/|\tau\eta^{(k)}|)$, which is much less than the total number of $\binom{n}{k}$ potential interactions of the $k$-th order in most cases.

In fact, the final number of valid interactions also depends on the value of $p$. Although we cannot theoretically guarantee different DNNs all have small $p$ values, experiments in Figure 5(b) show that $p$ was around 0.9 to 1.5 on common tasks. Nevertheless, Eq. (6) just shows an upper bound of valid interactions, which is much more than the real interaction number. Please see Table 2 for the higher sparsity of real interactions than the upper bound. Thus, Theorem 3 shows that it is quite common for a DNN to encode sparse interactions.

• **Case 2: When positive and negative interactions of the $k$-th order almost cancel each other.** In this case, the absolute value of $\eta^{(k)}$ can be extremely small. In such an extreme situation, the number of valid interactions is proportional to $n^{p+\delta}/|\tau\eta^{(k)}|$. Then, the upper bound for the number of interactions $R^{(k)}$ is much higher, according to Theorem 3. However, $R^{(k)}$ is still much less than $\binom{n}{k}$ if $|\eta^{(k)}|$ is not exponentially small.

**Real number of valid interactions vs. the derived bound.** We conducted experiments to compare the real number of valid interactions with the derived upper bound. We used the same LLMs and dataset as those in previous experiments. Besides, we also trained a 5-layer multi-layer perceptron (MLP) on a tabular dataset named *TV news* (Dua & Graff, 2017). Specifically, we set $M = 9$. We set $\tau = 0.05 \max_{S'} |I(S')|$ for all LLMs, and set $\tau = 0.1 \max_{S'} |I(S')|$ for the MLP. Table 2 shows that the real number of valid interactions was about 30 to 50, while the derived bound was about 200 to 260. Note that the number of all potential interactions was $2^n = 2^{10} = 1024$. Please see Appendix E.2 for the number of valid interactions and the bound on several example sentences.

### 3.3 WHEN ARE INTERACTIONS NON-SPARSE?

Despite the above proof of the sparsity of interactions under certain assumptions, there exist some special cases in which the DNN does not encode sparse interactions.

**Scenario 1.** Let us consider the scenario where the network output contains some random noise, *i.e.*, we can decompose the network output into $v'(\boldsymbol{x}_S) = v(\boldsymbol{x}_S) + \epsilon_S$, where $\epsilon_S$ denotes a fully random noise. In this case, the effect of each interaction can be rewritten as $I'(S) = I(S) + I_\epsilon(S)$, where $I(S)$ denotes the interaction extracted from the normal output component $v(\boldsymbol{x}_S)$, and $I_\epsilon(S) = \sum_{T \subseteq S}(-1)^{|S|-|T|} \cdot (\epsilon_T - \epsilon_\emptyset)$ denotes the interaction extracted from the noise. Because $I_\epsilon(S)$ is a sum of $2^{|S|}$ noise terms, the variance of $I_\epsilon(S)$ *w.r.t.* the noise is $2^{|S|}$ times larger. In this case, the value of $I_\epsilon(S)$, as well as $I'(S)$, is not likely to be zero, *i.e.*, we will not obtain sparse interactions.

In fact, this scenario does not satisfy Assumptions 1-$\alpha$ and 1-$\beta$. In this case, we can simply ignore small noises in $v'(\boldsymbol{x}_S)$ to obtain sparse interactions. To this end, Li & Zhang (2023a) proposed a method to estimate and remove potential small noises from the network output and boost the sparsity.

**Scenario 2.** Let us consider the second scenario, in which the network output on a masked sample $\boldsymbol{x}_S$ is purely dependent on the number of variables in $S$. We consider an example in which the sign of $u(S)$ is decided by the parity of the number of variables in $S$, *i.e.*, if $|S|$ is odd, then $u(S) = +1$; otherwise, $u(S) = -1$. In this case, $I(S)$ is always positive if $|S|$ is odd, and always negative if $|S|$ is even, which is not sparse. In fact, this scenario does not satisfy Assumption 2 (the monotonicity assumption). In addition, the classification of the parity does not represent the typical paradigm for classification because there are no inference patterns for classification.

**Scenario 3.** In this scenario, the network encodes high-order OR relationships between the input variables. For example, the network may encode an OR relationship *"blue patch 1"* $\vee \cdots \vee$ *"blue patch m"* to recognize the sky. Such an OR relationship will be explained as a large number of lower-order Harsanyi interactions. However, in real applications, such high-order interactions, *e.g.*, the detection of the sky, can actually be taken as a single concept. If we disentangle and remove such high-order interactions from the network output, the remaining output is likely to generate sparse Harsanyi interactions. In addition, this scenario does not satisfy Assumptions 1-$\alpha$ and 1-$\beta$.

**Scenario 4.** Let us consider the scenario in which the network output is in the form of a periodic function, *e.g.*, $v(\boldsymbol{x}_S) = \sin(\sum_{i \in S} x_i)$. In this case, interactions encoded by the network are not sparse. This scenario does not satisfy Assumptions 1-$\alpha$, 1-$\beta$, and Assumption 2.

**Scenario 5.** Let us consider the special task that heavily relies on the information of all input variables, *e.g.*, judging whether the number of 1's in a binary sequence (*e.g.*, the sequence $[0, 0, 1, 0, 1]$) is odd or even. In this task, the value of $p$ may be large, and the monotonicity assumption is not satisfied. As a result, the interactions may not be sparse. See Appendix E.3 for experiments.

## 4 SIGNIFICANCE OF THE EMERGENCE OF INTERACTION PRIMITIVES

• The emergence of symbolic interaction primitives can be considered as a theoretical foundation for the field of explainable AI. Many studies have attempted to explain the feature representation of DNNs as the encoding of different concepts with clear meanings. For example, Bau et al. (2017; 2020) examined how each convolutional filter was related to its encoded fuzzy concept. Kim et al. (2018) attempted to find a certain feature direction in the intermediate layer corresponding to a specific manually-defined concept. However, the fundamental issue behind these explanations, *i.e.*, whether a DNN can be faithfully explained as symbolic concepts, has not been studied.

• The proof of the emergence of interaction primitives may provide a new perspective to analyze the generalization power and robustness of a DNN. In fact, game-theoretic interactions have been used to explain overfitting (Zhang et al., 2020), adversarial robustness (Ren et al., 2021), and adversarial transferability (Wang et al., 2021), although these studies used multi-order interactions (Zhang et al., 2020) for analysis[5], rather than the Harsanyi interaction.

• Given the sparsity property and the universal matching property, the sample-wise transferability property can be obtained by contradiction. If interactions are not transferable across samples, then it means that the DNN activates different sets of salient interactions on different samples, and this indicates that the number of interaction patterns encoded by the DNN will be explosively large.

## 5 CONCLUSION AND FUTURE WORK

This study provided a set of conditions for the emergence of symbolic interaction primitives for inference. Under these conditions, we proved that a DNN encoded very sparse interactions between input variables. The commonness of these conditions indicated that the emergence of symbolic (sparse) interactions was not unusual, but might be a universal phenomenon for various well-trained AI models, because the proof does not rely on the network architecture. In future work, we will attempt to derive a tighter upper bound for the number of valid interactions $R^{(k)}$ encoded by the network. Additionally, we will further quantify the precise conditions that prevent the emergence of symbolic interaction primitives from occurring in a DNN.

---

[5]Harsanyi interactions are compositional elements of multi-order interactions (Ren et al., 2023a).

**Acknowledgements.** This work is partially supported by the National Science and Technology Major Project (2021ZD0111602), the National Nature Science Foundation of China (92370115, 62276165, 62206170).

## ETHIC STATEMENT

This paper aims to prove the emergence of sparse primitive inference patterns in well-trained DNNs under certain conditions. The proof potentially provides theoretical support for explaining DNNs at the concept level. There are no ethical issues with this paper.

## REPRODUCIBILITY STATEMENT

We have provided proofs for the theoretical results of this study in Appendix B. We have also provided experimental details in Appendix C.1, Appendix C.2, and Appendix D.1. Furthermore, we have released the code at `https://github.com/sjtu-xai-lab/interaction-sparsity`.

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

# A AXIOMS AND THEOREMS FOR THE HARSANYI DIVIDEND INTERACTION

## A.1 DESIRABLE GAME-THEORETIC PROPERTIES

The Harsanyi dividend was designed as a standard metric to measure interactions between input variables encoded by the network. In this section, we present several desirable axioms and theorems that the Harsanyi dividend interaction $I(S)$ satisfies. This further demonstrates the trustworthiness of the Harsanyi dividend interaction.

The Harsanyi dividend interactions $I(S)$ satisfies the *efficiency, linearity, dummy, symmetry, anonymity, recursive* and *interaction distribution* axioms, as follows. We follow the notation in the main paper to let $u(S) = v(\boldsymbol{x}_S) - v(\boldsymbol{x}_\emptyset)$.

(1) *Efficiency axiom* (proven by Harsanyi (1963)). The output score of a model can be decomposed into interaction effects of different patterns, *i.e.* $u(N) = \sum_{S \subseteq N} I(S)$.

(2) *Linearity axiom.* If we merge output scores of two models $u_1$ and $u_2$ as the output of model $u$, *i.e.* $\forall S \subseteq N, u(S) = u_1(S) + u_2(S)$, then their interaction effects $I_{u_1}(S)$ and $I_{u_2}(S)$ can also be merged as $\forall S \subseteq N, I_u(S) = I_{u_1}(S) + I_{u_2}(S)$.

(3) *Dummy axiom.* If a variable $i \in N$ is a dummy variable, *i.e.* $\forall S \subseteq N\backslash\{i\}, u(S \cup \{i\}) = u(S) + u(\{i\})$, then it has no interaction with other variables, $\forall \emptyset \neq S \subseteq N\backslash\{i\}, I(S \cup \{i\}) = 0$.

(4) *Symmetry axiom.* If input variables $i, j \in N$ cooperate with other variables in the same way, $\forall S \subseteq N\backslash\{i,j\}, u(S\cup\{i\}) = u(S\cup\{j\})$, then they have same interaction effects with other variables, $\forall S \subseteq N\backslash\{i,j\}, I(S \cup \{i\}) = I(S \cup \{j\})$.

(5) *Anonymity axiom.* For any permutations $\pi$ on $N$, we have $\forall S \subseteq N, I_u(S) = I_{\pi u}(\pi S)$, where $\pi S \triangleq \{\pi(i) | i \in S\}$, and the new model $\pi u$ is defined by $(\pi u)(\pi S) = u(S)$. This indicates that interaction effects are not changed by permutation.

(6) *Recursive axiom.* The interaction effects can be computed recursively. For $i \in N$ and $S \subseteq N\backslash\{i\}$, the interaction effect of the pattern $S \cup \{i\}$ is equal to the interaction effect of $S$ with the presence of $i$ minus the interaction effect of $S$ with the absence of $i$, *i.e.* $\forall S \subseteq N \backslash \{i\}, I(S \cup \{i\}) = I(S|i \text{ is always present}) - I(S)$. $I(S|i \text{ is always present})$ denotes the interaction effect when the variable $i$ is always present as a constant context, *i.e.* $I(S|i \text{ is always present}) = \sum_{L \subseteq S}(-1)^{|S|-|L|} \cdot u(L \cup \{i\})$.

(7) *Interaction distribution axiom.* This axiom characterizes how interactions are distributed for "interaction functions" (Sundararajan et al., 2020). An interaction function $u_T$ parameterized by a subset of variables $T$ is defined as follows. $\forall S \subseteq N$, if $T \subseteq S$, $u_T(S) = c$; otherwise, $u_T(S) = 0$. The function $u_T$ models pure interaction among the variables in $T$, because only if all variables in $T$ are present, the output value will be increased by $c$. The interactions encoded in the function $u_T$ satisfies $I(T) = c$, and $\forall S \neq T, I(S) = 0$.

## A.2 CONNECTION TO EXISTING GAME-THEORETIC ATTRIBUTION/INTERACTION METRICS

The Harsanyi interaction $I(S)$ is actually the elementary mechanism of existing game-theoretic attribution/interaction metrics, as follows.

**Theorem 4** (Connection to the Shapley value (Shapley, 1953), proven in both Harsanyi (1963) and Appendix B.5). *Let $\phi(i)$ denote the Shapley value of an input variable $i$. Then, the Shapley value $\phi(i)$ can be explained as the result of uniformly assigning attributions of each Harsanyi interaction to each involving variable $i$, i.e., $\phi(i) = \sum_{S \subseteq N\backslash\{i\}} \frac{1}{|S|+1} I(S \cup \{i\})$.*

**Theorem 5** (Connection to the Shapley interaction index (Grabisch & Roubens, 1999), proven in both Ren et al. (2023a) and Appendix B.6). *Given a subset of input variables $T \subseteq N$, the Shapley interaction index $I^{Shapley}(T)$ can be represented as $I^{Shapley}(T) = \sum_{S \subseteq N\backslash T} \frac{1}{|S|+1} I(S \cup T)$. In other words, the index $I^{Shapley}(T)$ can be explained as uniformly allocating $I(S')$ s.t. $S' = S \cup T$ to the compositional variables of $S'$ if we treat the coalition of variables in $T$ as a single variable.*

**Theorem 6** (Connection to the Shapley Taylor interaction index (Sundararajan et al., 2020), proven in both Ren et al. (2023a) and Appendix B.7). *Given a subset of input variables $T \subseteq N$, the $k$-th order Shapley Taylor interaction index $I^{Shapley\text{-}Taylor}(T)$ can be represented as the weighted sum of interaction effects, i.e., $I^{Shapley\text{-}Taylor}(T) = I(T)$ if $|T| < k$; $I^{Shapley\text{-}Taylor}(T) = \sum_{S \subseteq N\backslash T} \binom{|S|+k}{k}^{-1} I(S \cup T)$ if $|T| = k$; and $I^{Shapley\text{-}Taylor}(T) = 0$ if $|T| > k$.*

## B PROOF OF THEOREMS

### B.1 PROOF OF THEOREM 1 IN THE MAIN PAPER

**Theorem 1.** *Let the input sample $x$ be arbitrarily masked to obtain a masked sample $x_S$. The output of the DNN on the masked sample $x_S$ can be disentangled into the sum of effects of all interactions within the set $S$:*

$$\forall S \subseteq N, \ v(\boldsymbol{x}_S) = \sum_{T \subseteq S} I(T) + v(\boldsymbol{x}_\emptyset). \tag{7}$$

*Proof.* According to the definition of the Harsanyi interaction, we have $\forall S \subseteq N$,

$$
\begin{aligned}
\sum_{T \subseteq S} I(T) &= \sum_{T \subseteq S} \sum_{L \subseteq T} (-1)^{|T|-|L|} u(L) \\
&= \sum_{L \subseteq S} \sum_{T \subseteq S : T \supseteq L} (-1)^{|T|-|L|} u(L) \\
&= \sum_{L \subseteq S} \sum_{t=|L|}^{|S|} \sum_{\substack{T \subseteq S : S \supseteq L \\ |T|=t}} (-1)^{t-|L|} u(L) \\
&= \sum_{L \subseteq S} u(L) \sum_{m=0}^{|S|-|L|} \binom{|S|-|L|}{m} (-1)^m \\
&= u(S) = v(\boldsymbol{x}_S) - v(\boldsymbol{x}_\emptyset).
\end{aligned}
$$

Therefore, we have $v(\boldsymbol{x}_S) = \sum_{T \subseteq S} I(T) + v(\boldsymbol{x}_\emptyset)$.

$\square$

### B.2 DERIVATION OF ASSUMPTION 1-$\alpha$ FROM ASSUMPTION 1-$\beta$ IN THE MAIN PAPER

Before we give the derivation of Assumption 1-$\alpha$ from Assumption 1-$\beta$, we first prove the following lemma.

**Lemma 1.** *The effect $I(S)$ $(S \neq \emptyset)$ of an interactive concept can be rewritten as*

$$I(S) = \sum_{\boldsymbol{\kappa} \in Q_S} \left. \frac{\partial^{\kappa_1 + \cdots + \kappa_n} v}{\partial x_1^{\kappa_1} \cdots \partial x_n^{\kappa_n}} \right|_{\boldsymbol{x}=\boldsymbol{b}} \cdot \frac{\prod_{i \in S}(x_i - b_i)^{\kappa_i}}{\kappa_1! \cdots \kappa_n!}, \tag{8}$$

*where $Q_S = \{[\kappa_1, \ldots, \kappa_n]^\top \mid \forall i \in S, \kappa_i \in \mathbb{N}^+; \forall i \notin S, \kappa_i = 0\}$.*

Note that a similar proof was first introduced in Ren et al. (2023c).

*Proof.* Let us denote the function on the right of Eq. (8) by $\tilde{I}(S)$, *i.e.*, for $S \neq \emptyset$,

$$\tilde{I}(S) \stackrel{\text{def}}{=} \sum_{\boldsymbol{\kappa} \in Q_S} \left. \frac{\partial^{\kappa_1 + \cdots + \kappa_n} v}{\partial x_1^{\kappa_1} \cdots \partial x_n^{\kappa_n}} \right|_{\boldsymbol{x}=\boldsymbol{b}} \cdot \frac{\prod_{i \in S}(x_i - b_i)^{\kappa_i}}{\kappa_1! \cdots \kappa_n!}. \tag{9}$$

According to Eq. (1), we define $\tilde{I}(\emptyset) = 0$. Actually, it has been proven in Grabisch & Roubens (1999) and Ren et al. (2023a) that the Harsanyi interaction $I(S)$ defined in Eq. (1) is the **unique** metric satisfying the universal matching property mentioned in the main paper, *i.e.*,

$$\forall S \subseteq N, \ v(\boldsymbol{x}_S) = \sum_{T \subseteq S} I(T) + v(\boldsymbol{x}_\emptyset). \tag{10}$$

Thus, as long as we can prove that $\tilde{I}(S)$ also satisfies the above universal matching property, we can obtain $\tilde{I}(S) = I(S)$.

To this end, we only need to prove $\tilde{I}(S)$ also satisfies the universal matching property in Eq. (10). Specifically, given an input sample $\boldsymbol{x} \in \mathbb{R}^n$, let us consider the Taylor expansion of the network

output $v(\boldsymbol{x}_S)$ of an arbitrarily masked sample $\boldsymbol{x}_S (S \subseteq N)$, which is expanded at $\boldsymbol{x}_\emptyset = \boldsymbol{b} = [b_1, \cdots, b_n]^\top$. Then, we have

$$\forall S \subseteq N, \quad v(\boldsymbol{x}_S) = \sum_{\kappa_1=0}^{\infty} \cdots \sum_{\kappa_n=0}^{\infty} \frac{\partial^{\kappa_1+\cdots+\kappa_n} v}{\partial x_1^{\kappa_1} \cdots \partial x_n^{\kappa_n}}\bigg|_{\boldsymbol{x}=\boldsymbol{b}} \cdot \frac{((\boldsymbol{x}_S)_1 - b_1)^{\kappa_1} \cdots ((\boldsymbol{x}_S)_n - b_n)^{\kappa_n}}{\kappa_1! \cdots \kappa_n!} \quad (11)$$

where $b_i$ denotes the baseline value to mask the input variable $x_i$.

According to the definition of the masked sample $\boldsymbol{x}_S$, we have that all variables in $S$ keep unchanged and other variables are masked to the baseline value. That is, $\forall i \in S, (\boldsymbol{x}_S)_i = x_i; \forall i \notin S, (\boldsymbol{x}_S)_i = b_i$. Hence, we obtain $\forall i \notin S, [(\boldsymbol{x}_S)_i - b_i]^{\kappa_i} = 0$. Then, among all Taylor expansion terms, only terms corresponding to degrees $\boldsymbol{\kappa}$ in the set $P_S = \{[\kappa_1, \cdots, \kappa_n]^\top \mid \forall i \in S, \kappa_i \in \mathbb{N}; \forall i \notin S, \kappa_i = 0\}$ may not be zero. Therefore, Eq. (11) can be re-written as

$$\forall S \subseteq N, \quad v(\boldsymbol{x}_S) = \sum_{\boldsymbol{\kappa} \in P_S} \frac{\partial^{\kappa_1+\cdots+\kappa_n} v}{\partial x_1^{\kappa_1} \cdots \partial x_n^{\kappa_n}}\bigg|_{\boldsymbol{x}=\boldsymbol{b}} \cdot \frac{\prod_{i \in S}(x_i - b_i)^{\kappa_i}}{\kappa_1! \cdots \kappa_n!}. \quad (12)$$

We find that the set $P_S$ can be divided into multiple disjoint sets as $P_S = \cup_{T \subseteq S} Q_T$, where $Q_T = \{[\kappa_1, \cdots, \kappa_n]^\top \mid \forall i \in T, \kappa_i \in \mathbb{N}^+; \forall i \notin T, \kappa_i = 0\}$. Then, we can derive that

$$\begin{aligned} \forall S \subseteq N, \quad v(\boldsymbol{x}_S) &= \sum_{T \subseteq S} \sum_{\boldsymbol{\kappa} \in Q_T} \frac{\partial^{\kappa_1+\cdots+\kappa_n} v}{\partial x_1^{\kappa_1} \cdots \partial x_n^{\kappa_n}}\bigg|_{\boldsymbol{x}=\boldsymbol{b}} \cdot \frac{\prod_{i \in T}(x_i - b_i)^{\kappa_i}}{\kappa_1! \cdots \kappa_n!} \\ &= \sum_{T \subseteq S} \tilde{I}(T) + v(\boldsymbol{x}_\emptyset). \end{aligned} \quad (13)$$

The last step is obtained as follows. When $T = \emptyset$, $Q_T$ only has one element $\boldsymbol{\kappa} = [0, \cdots, 0]^\top$, which corresponds to the term $v(\boldsymbol{x}_\emptyset)$. Also, $\tilde{I}(\emptyset) = 0$, which leads to the final form. Thus, $\tilde{I}(S)$ satisfies the universal matching property in Eq. (10), and this lemma holds.

$\square$

Then, we prove that by combining Lemma 1 and Assumption 1-$\beta$, we can obtain Assumption 1-$\alpha$.

**Assumption 1-$\alpha$.** *Interactions of higher than the $M$-th order have zero effect, i.e., $\forall S \in \{S \subseteq N \mid |S| \geq M+1\}$, $I(S) = 0$. Here, the order of an interaction is defined as the number of input variables in $S$, i.e., $\mathrm{order}(S) \stackrel{\mathrm{def}}{=} |S|$.*

*Proof.* According to Lemma 1, we have

$$I(S) = \sum_{\boldsymbol{\kappa} \in Q_S} \frac{\partial^{\kappa_1+\cdots+\kappa_n} v}{\partial x_1^{\kappa_1} \cdots \partial x_n^{\kappa_n}}\bigg|_{\boldsymbol{x}=\boldsymbol{b}} \cdot \frac{\prod_{i \in S}(x_i - b_i)^{\kappa_i}}{\kappa_1! \cdots \kappa_n!}, \quad (14)$$

where $Q_S = \{[\kappa_1, \ldots, \kappa_n]^\top \mid \forall i \in S, \kappa_i \in \mathbb{N}^+; \forall i \notin S, \kappa_i = 0\}$.

We note that when $|S| \geq M+1$, we have $\forall \boldsymbol{\kappa} \in Q_S$, $\kappa_1 + \cdots + \kappa_n \geq M+1$. Then, combining with Assumption 1-$\beta$, we have

$$\frac{\partial^{\kappa_1+\cdots+\kappa_n} v}{\partial x_1^{\kappa_1} \cdots \partial x_n^{\kappa_n}}\bigg|_{\boldsymbol{x}=\boldsymbol{b}} = 0, \quad \forall \boldsymbol{\kappa} \in Q_S. \quad (15)$$

This leads to $I(S) = 0$, $\forall S \in \{S \subseteq N \mid |S| \geq M+1\}$. $\square$

### B.3 PROOF OF THEOREM 2 IN THE MAIN PAPER

To facilitate the proof of Theorem 2 in the main paper, we first prove the following two lemmas.

**Lemma 2.** *For each $M \leq m \leq n$, the average network output of the $m$-th order can be written as*

$$\bar{u}^{(m)} = \sum_{k=1}^{M} \frac{\binom{m}{k}}{\binom{n}{k}} A^{(k)}, \quad (16)$$

*where $A^{(k)} = \sum_{T \subseteq N, |T|=k} I(T)$.*

*Proof.* According to the definition of $\bar{u}^{(m)}$, we have

$$\bar{u}^{(m)} = \mathbb{E}_{S \subseteq N, |S|=m}[u(S)] \tag{17}$$

$$= \mathbb{E}_{S \subseteq N, |S|=m}[v(\boldsymbol{x}_S) - v(\boldsymbol{x}_\emptyset)] \tag{18}$$

$$= \mathbb{E}_{S \subseteq N, |S|=m}\left[\sum_{T \subseteq S} I(T)\right] \quad //\text{according to Theorem 1} \tag{19}$$

$$= \mathbb{E}_{S \subseteq N, |S|=m}\left[\sum_{k=1}^{m} \sum_{T \subseteq S, |T|=k} I(T)\right] \tag{20}$$

$$= \sum_{k=1}^{m} \mathbb{E}_{S \subseteq N, |S|=m}\left[\sum_{T \subseteq S, |T|=k} I(T)\right] \tag{21}$$

$$= \sum_{k=1}^{m} \frac{1}{\binom{n}{m}} \sum_{S \subseteq N, |S|=m} \sum_{T \subseteq S, |T|=k} I(T) \tag{22}$$

$$= \sum_{k=1}^{m} \frac{1}{\binom{n}{m}} \binom{n-k}{m-k} \sum_{T \subseteq N, |T|=k} I(T) \tag{23}$$

$$= \sum_{k=1}^{m} \frac{\binom{m}{k}}{\binom{n}{k}} \sum_{T \subseteq N, |T|=k} I(T) \tag{24}$$

$$= \sum_{k=1}^{m} \frac{\binom{m}{k}}{\binom{n}{k}} A^{(k)} \tag{25}$$

From Eq. (22) to Eq. (23), we note that each single $T \subseteq N(|T| = k)$ is repeatly counted for $\binom{n-k}{m-k}$ times in the sum $\sum_{S \subseteq N, |S|=m} \sum_{T \subseteq S, |T|=k} I(T)$. And from Eq. (23) to Eq. (24), we use the following property:

$$\binom{n}{m}\binom{m}{k} = \frac{n!}{m!(n-m)!}\frac{m!}{k!(m-k)!} = \frac{n!}{k!(n-k)!}\frac{(n-k)!}{(m-k)!(n-m)!} = \binom{n}{k}\binom{n-k}{m-k}. \tag{26}$$

Furthermore, according to Assumption 1-$\alpha$, we have $\forall S \in \{S \subseteq N \mid |S| \geq M+1\}$, $I(S) = 0$. Therefore, we have $\forall k \geq M+1$, $A^{(k)} = 0$. This leads to

$$\forall M \leq m \leq n, \quad \bar{u}^{(m)} = \sum_{k=1}^{m} \frac{\binom{m}{k}}{\binom{n}{k}} A^{(k)} = \sum_{k=1}^{M} \frac{\binom{m}{k}}{\binom{n}{k}} A^{(k)}. \tag{27}$$

$\square$

**Lemma 3.** *Given* $n \in \mathbb{N}^+$ *and* $M \in \mathbb{N}^+$, *where* $M < n$, *if* $\forall m \in \{n, n-1, \cdots, n-M\}$, $\sum_{k=1}^{M} \frac{\binom{m}{k}}{\binom{n}{k}} w_k = 0$, *then we have* $w_k = 0$ *for* $k = 1, \cdots, M$.

*Proof.* We first represent the above problem using the matrix representation: if $\boldsymbol{Cw} = \boldsymbol{0}$, where

$$\boldsymbol{C} = \begin{bmatrix} \frac{\binom{n}{1}}{\binom{n}{1}} & \frac{\binom{n}{2}}{\binom{n}{2}} & \cdots & \frac{\binom{n}{M}}{\binom{n}{M}} \\ \frac{\binom{n-1}{1}}{\binom{n}{1}} & \frac{\binom{n-1}{2}}{\binom{n}{2}} & \cdots & \frac{\binom{n-1}{M}}{\binom{n}{M}} \\ \vdots & \vdots & \ddots & \vdots \\ \frac{\binom{n-M}{1}}{\binom{n}{1}} & \frac{\binom{n-M}{2}}{\binom{n}{2}} & \cdots & \frac{\binom{n-M}{M}}{\binom{n}{M}} \end{bmatrix} \in \mathbb{R}^{(M+1)\times M}, \quad \text{and} \quad \boldsymbol{w} = \begin{bmatrix} w_1 \\ w_2 \\ \vdots \\ w_M \end{bmatrix} \in \mathbb{R}^M, \tag{28}$$

then we have $\boldsymbol{w} = 0$.

To prove this lemma, we only need to prove that $\text{rank}(\boldsymbol{C}) = M$.

If we perform elementary row transformations on the matrix, *i.e.*, subtracting the $(i+1)$-th row from the $i$-th row ($i = 1, 2, \cdots, M$), and using the formula $\binom{n}{m} - \binom{n-1}{m} = \binom{n-1}{m-1}$, the matrix can be

transformed as below, with its row rank unchanged:

$$
\boldsymbol{C'} = \begin{bmatrix}
\frac{\binom{n-1}{0}}{\binom{n}{1}} & \frac{\binom{n-1}{1}}{\binom{n}{2}} & \cdots & \frac{\binom{n-1}{M-1}}{\binom{n}{M}} \\
\frac{\binom{n-2}{0}}{\binom{n}{1}} & \frac{\binom{n-2}{1}}{\binom{n}{2}} & \cdots & \frac{\binom{n-2}{M-1}}{\binom{n}{M}} \\
\vdots & \vdots & \ddots & \vdots \\
\frac{\binom{n-M}{0}}{\binom{n}{1}} & \frac{\binom{n-M}{1}}{\binom{n}{2}} & \cdots & \frac{\binom{n-M}{M-1}}{\binom{n}{M}} \\
\frac{\binom{n-M}{1}}{\binom{n}{1}} & \frac{\binom{n-M}{2}}{\binom{n}{2}} & \cdots & \frac{\binom{n-M}{M}}{\binom{n}{M}}
\end{bmatrix} \tag{29}
$$

To prove that its rank is $M$, we only need to prove that the first $M$ rows of $\boldsymbol{C'}$ are linearly independent, which is further equivalent to proving a non-zero determinant of the square matrix consisting of the first $M$ rows, *i.e.*,

$$
D \stackrel{\text{def}}{=} \det \begin{bmatrix}
\frac{\binom{n-1}{0}}{\binom{n}{1}} & \frac{\binom{n-1}{1}}{\binom{n}{2}} & \cdots & \frac{\binom{n-1}{M-1}}{\binom{n}{M}} \\
\frac{\binom{n-2}{0}}{\binom{n}{1}} & \frac{\binom{n-2}{1}}{\binom{n}{2}} & \cdots & \frac{\binom{n-2}{M-1}}{\binom{n}{M}} \\
\vdots & \vdots & \ddots & \vdots \\
\frac{\binom{n-M}{0}}{\binom{n}{1}} & \frac{\binom{n-M}{1}}{\binom{n}{2}} & \cdots & \frac{\binom{n-M}{M-1}}{\binom{n}{M}}
\end{bmatrix} \neq 0 \tag{30}
$$

We can recursively obtain the equations below:

$$
D \prod_{k=1}^{M} \binom{n}{k} = \det \begin{bmatrix}
\binom{n-1}{0} & \binom{n-1}{1} & \cdots & \binom{n-1}{M-1} \\
\binom{n-2}{0} & \binom{n-2}{1} & \cdots & \binom{n-2}{M-1} \\
\vdots & \vdots & \ddots & \vdots \\
\binom{n-M}{0} & \binom{n-M}{1} & \cdots & \binom{n-M}{M-1}
\end{bmatrix} \tag{31}
$$

$$
= \det \begin{bmatrix}
0 & \binom{n-2}{0} & \cdots & \binom{n-2}{M-2} \\
0 & \binom{n-3}{0} & \cdots & \binom{n-3}{M-2} \\
\vdots & \vdots & \ddots & \vdots \\
\binom{n-M}{0} & \binom{n-M}{1} & \cdots & \binom{n-M}{M-1}
\end{bmatrix} \quad \text{// subtracting } (i+1)\text{-th row from } i\text{-th row} \tag{32}
$$

$$
= \det \begin{bmatrix}
\binom{n-2}{0} & \binom{n-2}{1} & \cdots & \binom{n-2}{M-2} \\
\binom{n-3}{0} & \binom{n-2}{1} & \cdots & \binom{n-2}{M-2} \\
\vdots & \vdots & \ddots & \vdots \\
\binom{n-M}{0} & \binom{n-M}{1} & \cdots & \binom{n-M}{M-2}
\end{bmatrix} \tag{33}
$$

$$
= \cdots \tag{34}
$$

$$
= 1 \tag{35}
$$

Now we can see that

$$
D = \frac{1}{\prod_{k=1}^{M} \binom{n}{k}} \neq 0 \tag{36}
$$

This leads to the conclusion:

If $\forall \, m \in \{n, n-1, \cdots, n-M\}$, $\sum_{k=1}^{M} \frac{\binom{m}{k}}{\binom{n}{k}} w_k = 0$, then $w_k = 0$ for $k = 1, \cdots, M$.

$\square$

Next, we will prove Theorem 2.

**Theorem 2.** *There exists $m_0 \in \{n, n-1, \cdots, n-M\}$, such that for all $1 \leq k \leq M$, the sum of the effects of all $k$-order interactions can be written as:*

$$
A^{(k)} = \left( \lambda^{(k)} n^{p+\delta} + a_{\lfloor p \rfloor - 1}^{(k)} n^{\lfloor p \rfloor - 1} + \cdots + a_1^{(k)} n + a_0^{(k)} \right) \bar{u}^{(1)}, \tag{37}
$$

*where $|\lambda^{(k)}| \leq 1$, $|a_0^{(k)}| < n$, $|a_i^{(k)}| \in \{0, 1, \cdots, n-1\}$ for $i = 1, \cdots, \lfloor p \rfloor - 1$, and*

$$\delta \leq \log_n \left( \frac{1}{\lambda} \left( 1 - \frac{a_{\lfloor p \rfloor - 1}}{n^{p - \lfloor p \rfloor + 1}} - \cdots - \frac{a_0}{n^p} \right) \right), \quad \text{if } \lambda > 0, \tag{38}$$

$$\delta \leq \log_n \left( \frac{1}{-\lambda} \left( \frac{a_{\lfloor p \rfloor - 1}}{n^{p - \lfloor p \rfloor + 1}} + \cdots + \frac{a_0}{n^p} \right) \right), \quad \text{if } \lambda < 0. \tag{39}$$

Here, $\lambda \overset{\text{def}}{=} \sum_{k=1}^{M} \frac{\binom{m_0}{k}}{\binom{n}{k}} \lambda^{(k)} \neq 0$, $a_i \overset{\text{def}}{=} \sum_{k=1}^{M} \frac{\binom{m_0}{k}}{\binom{n}{k}} a_i^{(k)}$ for $i = 0, 1, \cdots, \lfloor p \rfloor - 1$, and $\lfloor p \rfloor$ denotes the greatest integer less than or equal to $p$.

*Proof.* First, according to Assumption 3, if we let $m' = 1$, then the average output of order $m$ should satisfy $\bar{u}^{(m)} \leq m^p \cdot \bar{u}^{(1)}$. Then, according to Assumption 2, we further have $\bar{u}^{(m)} \geq \bar{u}^{(0)} = v(\boldsymbol{x}_\emptyset) - v(\boldsymbol{x}_\emptyset) = 0$. Combining with the conclusion in Lemma 2, we have

$$\forall M \leq m \leq n, \quad 0 \leq \bar{u}^{(m)} = \sum_{k=1}^{M} \frac{\binom{m}{k}}{\binom{n}{k}} A^{(k)} \leq m^p \cdot \bar{u}^{(1)}. \tag{40}$$

**Deriving the form of $A^{(k)}$.** For each $A^{(k)}$, $1 \leq k \leq M$, we consider its $n$-ary representation after being divided by $\bar{u}^{(1)}$, *i.e.*,

$$\frac{A^{(k)}}{\bar{u}^{(1)}} = a_{q^{(k)}}^{(k)} n^{q^{(k)}} + a_{q^{(k)} - 1}^{(k)} n^{q^{(k)} - 1} + \cdots + a_1^{(k)} n + a_0^{(k)}, \tag{41}$$

$$\Rightarrow A^{(k)} = \left( a_{q^{(k)}}^{(k)} n^{q^{(k)}} + a_{q^{(k)} - 1}^{(k)} n^{q^{(k)} - 1} + \cdots + a_1^{(k)} n + a_0^{(k)} \right) \bar{u}^{(1)}, \tag{42}$$

where $q^{(k)} \in \mathbb{N}$, $|a_i^{(k)}| \in \{0, 1, \cdots, n-1\}$ for $i = 1, \cdots, q^{(k)}$, and $|a_0^{(k)}| < n$. Note that we absorb the decimal part into $a_0^{(k)}$. Then, let us consider the following two cases for each $1 \leq k \leq M$. We will show that in both cases, $A^{(k)}$ can be represented as the form in Eq. (37).

**Case 1**: if $q^{(k)} \leq \lfloor p \rfloor - 1$. In this case, $A^{(k)}$ can directly be represented as the form in Eq. (37):

$$A^{(k)} = \left( \lambda^{(k)} n^{p+\delta} + a_{\lfloor p \rfloor - 1}^{(k)} n^{\lfloor p \rfloor - 1} + \cdots + a_{q^{(k)} + 1}^{(k)} n^{q^{(k)} + 1} + a_{q^{(k)}}^{(k)} n^{q^{(k)}} + a_1^{(k)} n + a_0^{(k)} \right) \bar{u}^{(1)}, \tag{43}$$

where we simply set $\lambda^{(k)} = a_{\lfloor p \rfloor - 1}^{(k)} = \cdots = a_{q^{(k)} + 1}^{(k)} = 0$, and $\delta$ can be arbitrary (we will discuss the choice and range of $\delta$ in Case 2).

**Case 2**: if $q^{(k)} \geq \lfloor p \rfloor$. In this case, we first merge all terms with a degree higher than or equal to $\lfloor p \rfloor$ into a single term. According to Eq. (42), we have

$$A^{(k)} = \left( \underbrace{a_{q^{(k)}}^{(k)} n^{q^{(k)}} + \cdots + a_{\lfloor p \rfloor}^{(k)} n^{\lfloor p \rfloor}}_{=: s^{(k)} n^{p+\delta^{(k)}}} + a_{\lfloor p \rfloor - 1}^{(k)} n^{\lfloor p \rfloor - 1} + \cdots + a_1^{(k)} n + a_0^{(k)} \right) \bar{u}^{(1)} \tag{44}$$

$$= \left( s^{(k)} n^{p+\delta^{(k)}} + a_{\lfloor p \rfloor - 1}^{(k)} n^{\lfloor p \rfloor - 1} + \cdots + a_1^{(k)} n + a_0^{(k)} \right) \bar{u}^{(1)}, \tag{45}$$

where $s^{(k)} \in \{-1, 1\}$ denotes the sign of this term.

Let us consider all orders in the set $G = \{k \mid 1 \leq k \leq M, q^{(k)} \geq \lfloor p \rfloor\}$. Each order $k$ has the corresponding $\delta^{(k)}$. We set $\delta = \max_{k \in G} \delta^{(k)}$, and let $k^* = \arg\max_{k \in G} \delta^{(k)}$. Then, for any $k \in G$, we can rewrite the first term in Eq. (45) as

$$s^{(k)} n^{p+\delta^{(k)}} = \underbrace{s^{(k)} n^{\delta^{(k)} - \delta}}_{=: \lambda^{(k)}} n^{p+\delta} = \lambda^{(k)} n^{p+\delta}. \tag{46}$$

Note that since $\delta = \max_{k \in G} \delta^{(k)} \geq \delta^{(k)}$ and $s^{(k)} \in \{-1, 1\}$, we have

$$|\lambda^{(k)}| = |s^{(k)} n^{\delta^{(k)} - \delta}| = |n^{\delta^{(k)} - \delta}| \leq 1. \tag{47}$$

In particular, $|\lambda^{(k^*)}| = 1$ because $\delta^{(k^*)} = \delta$. Therefore, for any $k \in G$, $A^{(k)}$ can be rewritten as

$$A^{(k)} = \left(\lambda^{(k)}n^{p+\delta} + a^{(k)}_{\lfloor p \rfloor - 1}n^{\lfloor p \rfloor - 1} + \cdots + a^{(k)}_1 n + a^{(k)}_0\right)\bar{u}^{(1)}, \tag{48}$$

where $|\lambda^{(k)}| \leq 1$, $|a^{(k)}_0| < n$, $|a^{(k)}_i| \in \{0, 1, \cdots, n-1\}$ for $i = 1, \cdots, \lfloor p \rfloor - 1$.

**Deriving the upper bound of $\delta$.** Next, we will derive the upper bound for $\delta$ by using the inequality in Eq. (40). Let us plug in the expression of $A^{(k)}$ into the Eq. (40), and obtain

$$\bar{u}^{(m)} = \left(\left(\sum_{k=1}^{M}\frac{\binom{m}{k}}{\binom{n}{k}}\lambda^{(k)}\right)n^{p+\delta} + \left(\sum_{k=1}^{M}\frac{\binom{m}{k}}{\binom{n}{k}}a^{(k)}_{\lfloor p \rfloor - 1}\right)n^{\lfloor p \rfloor - 1} + \cdots + \left(\sum_{k=1}^{M}\frac{\binom{m}{k}}{\binom{n}{k}}a^{(k)}_0\right)\right)\bar{u}^{(1)} \tag{49}$$

According to Lemma 3, we have the following assertion:

$$\exists\, m_0 \in \{n, n-1, \cdots, n-M\}, \quad \sum_{k=1}^{M}\frac{\binom{m_0}{k}}{\binom{n}{k}}\lambda^{(k)} \neq 0 \tag{50}$$

This can be proven by contradiction: if for any $m \in \{n, n-1, \cdots, n-M\}$, we all have $\sum_{k=1}^{M}\frac{\binom{m_0}{k}}{\binom{n}{k}}\lambda^{(k)} = 0$, then according to Lemma 3, we obtain $\lambda^{(k)} = 0$ for all $1 \leq k \leq M$. However, we have mentioned above that $|\lambda^{(k^*)}| = 1$, which leads to contradiction.

Let us denote $\lambda \overset{\text{def}}{=} \sum_{k=1}^{M}\frac{\binom{m_0}{k}}{\binom{n}{k}}\lambda^{(k)} \neq 0$, $a_i \overset{\text{def}}{=} \sum_{k=1}^{M}\frac{\binom{m_0}{k}}{\binom{n}{k}}a^{(k)}_i$ for $i = 0, 1, \cdots, \lfloor p \rfloor - 1$, therefore using the inequality in Eq. (40) for $m = m_0$ we can write

$$0 \leq \bar{u}^{(m_0)} = \left(\lambda n^{p+\delta} + a_{\lfloor p \rfloor - 1}n^{\lfloor p \rfloor - 1} + \cdots + a_1 n + a_0\right)\bar{u}^{(1)} \leq m_0^p \cdot \bar{u}^{(1)} \leq n^p \cdot \bar{u}^{(1)}. \tag{51}$$

When $\lambda > 0$, by using the right-side inequality, we obtain

$$\delta \leq \log_n\left(\frac{1}{\lambda}\left(1 - \frac{a_{\lfloor p \rfloor - 1}}{n^{p - \lfloor p \rfloor + 1}} - \cdots - \frac{a_0}{n^p}\right)\right). \tag{52}$$

Similarly, when $\lambda < 0$, by using the left-side inequality, we obtain

$$\delta \leq \log_n\left(\frac{1}{-\lambda}\left(\frac{a_{\lfloor p \rfloor - 1}}{n^{p - \lfloor p \rfloor + 1}} + \cdots + \frac{a_0}{n^p}\right)\right). \tag{53}$$

$\square$

## B.4 Proof of Theorem 3 in the main paper

**Theorem 6.** $R^{(k)}$ *has the following upper bound:*

$$R^{(k)} \leq \frac{\bar{u}^{(1)}}{\tau|\eta^{(k)}|}|\lambda^{(k)}n^{p+\delta} + a^{(k)}_{\lfloor p \rfloor - 1}n^{\lfloor p \rfloor - 1} + \cdots + a^{(k)}_0|. \tag{54}$$

*Proof.* According to the definition of $A^{(k)}$, we have

$$A^{(k)} = \sum_{|S|=k} I(S) \tag{55}$$

$$= \eta^{(k)}\sum_{|S|=k}|I(S)| \quad // \text{ according to the definition of } \eta^{(k)} \tag{56}$$

Then, we obtain

$$\frac{A^{(k)}}{\eta^{(k)}} = \sum_{|S|=k}|I(S)| \tag{57}$$

$$\geq \sum_{|S|=k, |I(S)|\geq\tau}|I(S)| \tag{58}$$

$$\geq \tau R^{(k)} \quad // \text{ according to } R^{(k)} = |\{S \subseteq N \mid |S| = k, |I(S)| \geq \tau\}| \tag{59}$$

Note that $A^{(k)}$ always has the same sign as $\eta^{(k)}$, making $\frac{A^{(k)}}{\eta^{(k)}} > 0$. Therefore, we can simply add absolute value to the left-hand side of the above equation and get $\frac{|A^{(k)}|}{|\eta^{(k)}|} \geq \tau R^{(k)}$.

Since $\tau > 0$, we can obtain

$$R^{(k)} \leq \frac{|A^{(k)}|}{\tau|\eta^{(k)}|} = \frac{\bar{u}^{(1)}}{\tau|\eta^{(k)}|}|\lambda^{(k)}n^{p+\delta} + a^{(k)}_{\lfloor p \rfloor - 1}n^{\lfloor p \rfloor - 1} + \cdots + a^{(k)}_0|. \tag{60}$$

$\square$

### B.5 Proof of Theorem 4 in the Appendix

Before proving Theorem 4, we first prove the following lemma, which can serve as the foundation for proofs of Theorem 4, 5, and 6.

**Lemma 4** (Connection to the marginal benefit). $\Delta u_T(S) = \sum_{L \subseteq T}(-1)^{|T|-|L|}u(L \cup S)$ *denotes the marginal benefit (Grabisch & Roubens, 1999) of variables in $T \subseteq N \setminus S$ given the environment $S$. We have proven that $\Delta u_T(S)$ can be decomposed into the sum of interaction utilities inside $T$ and sub-environments of $S$, i.e. $\Delta u_T(S) = \sum_{S' \subseteq S} I(T \cup S')$.*

*Proof.* By the definition of the marginal benefit, we have

$$\Delta u_T(S) = \sum_{L \subseteq T}(-1)^{|T|-|L|}u(L \cup S)$$

$$= \sum_{L \subseteq T}(-1)^{|T|-|L|}\sum_{K \subseteq L \cup S}I(K) \quad \text{// by the universal matching property}$$

$$= \sum_{L \subseteq T}(-1)^{|T|-|L|}\sum_{L' \subseteq L}\sum_{S' \subseteq S}I(L' \cup S') \quad \text{// since } L \cap S = \emptyset$$

$$= \sum_{S' \subseteq S}\left[\sum_{L \subseteq T}(-1)^{|T|-|L|}\sum_{L' \subseteq L}I(L' \cup S')\right]$$

$$= \sum_{S' \subseteq S}\left[\sum_{L' \subseteq T}\sum_{\substack{L \subseteq T \\ L \supseteq L'}}(-1)^{|T|-|L|}I(L' \cup S')\right]$$

$$= \sum_{S' \subseteq S}\left[\underbrace{I(S' \cup T)}_{L'=T} + \underbrace{\sum_{L' \subsetneq T}\left(\sum_{l=|L'|}^{|T|}\binom{|T|-|L'|}{l-|L'|}(-1)^{|T|-|L|}I(L' \cup S')\right)}_{L' \subsetneq T}\right]$$

$$= \sum_{S' \subseteq S}\left[I(S' \cup T) + \sum_{L' \subsetneq T}\left(I(L' \cup S') \cdot \underbrace{\sum_{l=|L'|}^{|T|}\binom{|T|-|L'|}{l-|L'|}(-1)^{|T|-|L|}}_{=0}\right)\right]$$

$$= \sum_{S' \subseteq S}I(S' \cup T)$$

$\square$

Then, let us prove Theorem 4.

**Theorem 4.** *Let $\phi(i)$ denote the Shapley value of an input variable $i$. Then, the Shapley value $\phi(i)$ can be explained as the result of uniformly assigning attributions of each Harsanyi interaction to each involving variable $i$, i.e., $\phi(i) = \sum_{S \subseteq N \setminus \{i\}} \frac{1}{|S|+1}I(S \cup \{i\})$.*

*Proof.* By the definition of the Shapley value, we have

$$
\phi(i) = \mathbb{E}_{m} \mathbb{E}_{\substack{S \subseteq N \setminus \{i\} \\ |S|=m}} [u(S \cup \{i\}) - u(S)]
$$

$$
= \frac{1}{|N|} \sum_{m=0}^{|N|-1} \frac{1}{\binom{|N|-1}{m}} \sum_{\substack{S \subseteq N \setminus \{i\} \\ |S|=m}} \Big[ u(S \cup \{i\}) - u(S) \Big]
$$

$$
= \frac{1}{|N|} \sum_{m=0}^{|N|-1} \frac{1}{\binom{|N|-1}{m}} \sum_{\substack{S \subseteq N \setminus \{i\} \\ |S|=m}} \Delta u_{\{i\}}(S)
$$

$$
= \frac{1}{|N|} \sum_{m=0}^{|N|-1} \frac{1}{\binom{|N|-1}{m}} \sum_{\substack{S \subseteq N \setminus \{i\} \\ |S|=m}} \left[ \sum_{L \subseteq S} I(L \cup \{i\}) \right] \quad \text{// by Lemma 4}
$$

$$
= \frac{1}{|N|} \sum_{L \subseteq N \setminus \{i\}} \sum_{m=0}^{|N|-1} \frac{1}{\binom{|N|-1}{m}} \sum_{\substack{S \subseteq N \setminus \{i\} \\ |S|=m \\ S \supseteq L}} I(L \cup \{i\})
$$

$$
= \frac{1}{|N|} \sum_{L \subseteq N \setminus \{i\}} \sum_{m=|L|}^{|N|-1} \frac{1}{\binom{|N|-1}{m}} \sum_{\substack{S \subseteq N \setminus \{i\} \\ |S|=m \\ S \supseteq L}} I(L \cup \{i\}) \quad \text{// since } S \supseteq L, |S| = m \geq |L|.
$$

$$
= \frac{1}{|N|} \sum_{L \subseteq N \setminus \{i\}} \sum_{m=|L|}^{|N|-1} \frac{1}{\binom{|N|-1}{m}} \cdot \binom{|N|-|L|-1}{m-|L|} I(L \cup \{i\})
$$

$$
= \frac{1}{|N|} \sum_{L \subseteq N \setminus \{i\}} I(L \cup \{i\}) \underbrace{\sum_{k=0}^{|N|-|L|-1} \frac{1}{\binom{|N|-1}{|L|+k}} \cdot \binom{|N|-|L|-1}{k}}_{w_L}
$$

Then, we leverage the following properties of combinatorial numbers and the Beta function to simplify the term $w_L = \sum_{k=0}^{|N|-|L|-1} \frac{1}{\binom{|N|-1}{|L|+k}} \cdot \binom{|N|-|L|-1}{k}$.

*(i) A property of combinitorial numbers.* $m \cdot \binom{n}{m} = n \cdot \binom{n-1}{m-1}$.

*(ii) The definition of the Beta function.* For $p, q > 0$, the Beta function is defined as $B(p,q) = \int_0^1 x^{p-1}(1-x)^{1-q}dx$.

*(iii) Connections between combinitorial numbers and the Beta function.*

○ When $p, q \in \mathbb{Z}^+$, we have $B(p,q) = \frac{1}{q \cdot \binom{p+q-1}{p-1}}$.

○ For $m, n \in \mathbb{Z}^+$ and $n > m$, we have $\binom{n}{m} = \frac{1}{m \cdot B(n-m+1, m)}$.

$$
w_L = \sum_{k=0}^{|N|-|L|-1} \frac{1}{\binom{|N|-1}{|L|+k}} \cdot \binom{|N|-|L|-1}{k}
$$

$$
= \sum_{k=0}^{|N|-|L|-1} \binom{|N|-|L|-1}{k} \cdot (|L|+k) \cdot B(|N|-|L|-k, |L|+k)
$$

$$
= \sum_{k=0}^{|N|-|L|-1} |L| \cdot \binom{|N|-|L|-1}{k} \cdot B(|N|-|L|-k, |L|+k) \quad \cdots \text{①}
$$

$$
+ \sum_{k=0}^{|N|-|L|-1} k \cdot \binom{|N|-|L|-1}{k} \cdot B(|N|-|L|-k, |L|+k) \quad \cdots \text{②}
$$

Then, we solve ① and ② respectively. For ①, we have

$$\textcircled{1} = \int_0^1 |L| \sum_{k=0}^{|N|-|L|-1} \binom{|N|-|L|-1}{k} \cdot x^{|N|-|L|-k-1} \cdot (1-x)^{|L|+k-1} \, dx$$

$$= \int_0^1 |L| \cdot \underbrace{\left[ \sum_{k=0}^{|N|-|L|-1} \binom{|N|-|L|-1}{k} \cdot x^{|N|-|L|-k-1} \cdot (1-x)^k \right]}_{=1} \cdot (1-x)^{|L|-1} \, dx$$

$$= \int_0^1 |L|(1-x)^{|L|-1} \, dx = 1$$

For $\textcircled{2}$, we have

$$\textcircled{2} = \sum_{k=1}^{|N|-|L|-1} (|N|-|L|-1) \cdot \binom{|N|-|L|-2}{k-1} \cdot B(|N|-|L|-k, |L|+k)$$

$$= (|N|-|L|-1) \sum_{k'=0}^{|N|-|L|-2} \binom{|N|-|L|-2}{k'} \cdot B(|N|-|L|-k'-1, |L|+k'+1)$$

$$= (|N|-|L|-1) \int_0^1 \sum_{k'=0}^{|N|-|L|-2} \binom{|N|-|L|-2}{k'} \cdot x^{|N|-|L|-k'-2} \cdot (1-x)^{|L|+k'} \, dx$$

$$= (|N|-|L|-1) \int_0^1 \underbrace{\left[ \sum_{k'=0}^{|N|-|L|-2} \binom{|N|-|L|-2}{k'} \cdot x^{|N|-|L|-k'-2} \cdot (1-x)^{k'} \right]}_{=1} \cdot (1-x)^{|L|} \, dx$$

$$= (|N|-|L|-1) \int_0^1 (1-x)^{|L|} \, dx = \frac{|N|-|L|-1}{|L|+1}$$

Hence, we have

$$w_L = \textcircled{1} + \textcircled{2} = 1 + \frac{|N|-|L|-1}{|L|+1} = \frac{|N|}{|L|+1}$$

Therefore, we proved $\phi(i) = \frac{1}{|N|} \sum_{S \subseteq N \setminus \{i\}} w_L \cdot I(L \cup \{i\}) = \sum_{S \subseteq N \setminus \{i\}} \frac{1}{|S|+1} I(S \cup \{i\})$.

$\square$

### B.6 PROOF OF THEOREM 5 IN THE APPENDIX

**Theorem 5.** *Given a subset of input variables $T \subseteq N$, the Shapley interaction index $I^{Shapley}(T)$ can be represented as $I^{Shapley}(T) = \sum_{S \subseteq N \setminus T} \frac{1}{|S|+1} I(S \cup T)$. In other words, the index $I^{Shapley}(T)$ can be explained as uniformly allocating $I(S')$ s.t. $S' = S \cup T$ to the compositional variables of $S'$, if we treat the coalition of variables in $T$ as a single variable.*

*Proof.* The Shapley interaction index (Grabisch & Roubens, 1999) is defined as $I^{Shapley}(T) = \sum_{S \subseteq N \setminus T} \frac{|S|!(|N|-|S|-|T|)!}{(|N|-|T|+1)!} \Delta u_T(S)$. Then, we have

$$I^{\text{Shapley}}(T) = \sum_{S \subseteq N \setminus T} \frac{|S|!(|N| - |S| - |T|)!}{(|N| - |T| + 1)!} \Delta u_T(S)$$

$$= \frac{1}{|N| - |T| + 1} \sum_{m=0}^{|N|-|T|} \frac{1}{\binom{|N|-|T|}{m}} \sum_{\substack{S \subseteq N \setminus T \\ |\bar{S}| = m}} \Delta u_T(S)$$

$$= \frac{1}{|N| - |T| + 1} \sum_{m=0}^{|N|-|T|} \frac{1}{\binom{|N|-|T|}{m}} \sum_{\substack{S \subseteq N \setminus T \\ |\bar{S}| = m}} \left[ \sum_{L \subseteq S} I(L \cup T) \right] \quad \text{// by Lemma 4}$$

$$= \frac{1}{|N| - |T| + 1} \sum_{L \subseteq N \setminus T} \sum_{m=|L|}^{|N|-|T|} \frac{1}{\binom{|N|-|T|}{m}} \sum_{\substack{S \subseteq N \setminus T \\ |\bar{S}| = m \\ S \supseteq L}} I(L \cup T)$$

$$= \frac{1}{|N| - |T| + 1} \sum_{L \subseteq N \setminus T} \sum_{m=|L|}^{|N|-|T|} \frac{1}{\binom{|N|-|T|}{m}} \binom{|N| - |L| - |T|}{m - |L|} I(L \cup T)$$

$$= \frac{1}{|N| - |T| + 1} \sum_{L \subseteq N \setminus T} I(L \cup T) \underbrace{\sum_{k=0}^{|N|-|L|-|T|} \frac{1}{\binom{|N|-|T|}{|L|+k}} \binom{|N| - |L| - |T|}{k}}_{w_L}$$

Similar to the proof of Theorem 4, we leverage the properties of combinatorial numbers and the Beta function to simplify $w_L$.

$$w_L = \sum_{k=0}^{|N|-|L|-|T|} \frac{1}{\binom{|N|-|T|}{|L|+k}} \binom{|N| - |L| - |T|}{k}$$

$$= \sum_{k=0}^{|N|-|L|-|T|} \binom{|N| - |L| - |T|}{k} \cdot \left( |L| + k \right) \cdot B\left( |N| - |L| - |T| - k + 1, |L| + k \right)$$

$$= \sum_{k=0}^{|N|-|L|-|T|} |L| \cdot \binom{|N| - |L| - |T|}{k} \cdot B\left( |N| - |L| - |T| - k + 1, |L| + k \right) \quad \cdots \textcircled{1}$$

$$+ \sum_{k=0}^{|N|-|L|-|T|} k \cdot \binom{|N| - |L| - |T|}{k} \cdot B\left( |N| - |L| - |T| - k + 1, |L| + k \right) \quad \cdots \textcircled{2}$$

Then, we solve $\textcircled{1}$ and $\textcircled{2}$ respectively. For $\textcircled{1}$, we have

$$\textcircled{1} = \int_0^1 |L| \sum_{k=0}^{|N|-|L|-|T|} \binom{|N| - |L| - |T|}{k} \cdot x^{|N|-|L|-|T|-k} \cdot (1 - x)^{|L|+k-1} \, dx$$

$$= \int_0^1 |L| \cdot \underbrace{\left[ \sum_{k=0}^{|N|-|L|-|T|} \binom{|N| - |L| - |T|}{k} \cdot x^{|N|-|L|-|T|-k} \cdot (1 - x)^{k} \right]}_{=1} \cdot (1 - x)^{|L|-1} \, dx$$

$$= \int_0^1 |L| \cdot (1 - x)^{|L|-1} \, dx = 1$$

For $\textcircled{2}$, we have

$$
\text{\textcircled{2}} = \sum_{k=1}^{|N|-|L|-|T|} (|N|-|L|-|T|) \binom{|N|-|L|-|T|-1}{k-1} \cdot B\Big(|N|-|L|-|T|-k+1, |L|+k\Big)
$$

$$
= (|N|-|L|-|T|) \sum_{k'=0}^{|N|-|L|-|T|-1} \binom{|N|-|L|-|T|-1}{k'} \cdot B\Big(|N|-|L|-|T|-k', |L|+k'+1\Big)
$$

$$
= (|N|-|L|-|T|) \int_0^1 \sum_{k'=0}^{|N|-|L|-|T|-1} \binom{|N|-|L|-|T|-1}{k'} \cdot x^{|N|-|L|-|T|-k'-1} \cdot (1-x)^{|L|+k'} \, dx
$$

$$
= (|N|-|L|-|T|) \int_0^1 \underbrace{\left[ \sum_{k'=0}^{|N|-|L|-|T|-1} \binom{|N|-|L|-|T|-1}{k'} \cdot x^{|N|-|L|-|T|-k'-1} \cdot (1-x)^{k'} \right]}_{=1} \cdot (1-x)^{|L|} \, dx
$$

$$
= (|N|-|L|-|T|) \int_0^1 (1-x)^{|L|} \, dx = \frac{|N|-|L|-|T|}{|L|+1}
$$

Hence, we have

$$
w_L = \text{\textcircled{1}} + \text{\textcircled{2}} = 1 + \frac{|N|-|L|-|T|}{|L|+1} = \frac{|N|-|T|+1}{|L|+1}
$$

Therefore, we proved that $I^{\text{Shapley}}(T) = \frac{1}{|N|-|T|+1} \sum_{L \subseteq N \setminus T} w_L \cdot I(L \cup T) = \sum_{L \subseteq N \setminus T} \frac{1}{|L|+1} I(L \cup T)$.

$\square$

### B.7 PROOF OF THEOREM 6 IN THE APPENDIX

**Theorem 6.** *Given a subset of input variables $T \subseteq N$, the $k$-th order Shapley Taylor interaction index $I^{\text{Shapley-Taylor}}(T)$ can be represented as weighted sum of interaction effects, i.e., $I^{\text{Shapley-Taylor}}(T) = I(T)$ if $|T| < k$; $I^{\text{Shapley-Taylor}}(T) = \sum_{S \subseteq N \setminus T} \binom{|S|+k}{k}^{-1} I(S \cup T)$ if $|T| = k$; and $I^{\text{Shapley-Taylor}}(T) = 0$ if $|T| > k$.*

*Proof.* By the definition of the Shapley Taylor interaction index,

$$
I^{\text{Shapley-Taylor}(k)}(T) = \begin{cases} \Delta u_T(\emptyset) & \text{if } |T| < k \\ \frac{k}{|N|} \sum_{S \subseteq N \setminus T} \frac{1}{\binom{|N|-1}{|S|}} \Delta u_T(S) & \text{if } |T| = k \\ 0 & \text{if } |T| > k \end{cases}
$$

When $|T| < k$, by the definition in Eq. (1) of the main paper, we have

$$
I^{\text{Shapley-Taylor}(k)}(T) = \Delta u_T(\emptyset) = \sum_{L \subseteq T} (-1)^{|T|-|L|} \cdot u(L) = I(T).
$$

When $|T| = k$, we have

$$
\begin{aligned}
I^{\text{Shapley-Taylor}(k)}(T) =& \frac{k}{|N|} \sum_{S \subseteq N \setminus T} \frac{1}{\binom{|N|-1}{|S|}} \cdot \Delta u_T(S) \\
=& \frac{k}{|N|} \sum_{m=0}^{|N|-k} \sum_{\substack{S \subseteq N \setminus T \\ |S|=m}} \frac{1}{\binom{|N|-1}{|S|}} \cdot \Delta u_T(S) \\
=& \frac{k}{|N|} \sum_{m=0}^{|N|-k} \sum_{\substack{S \subseteq N \setminus T \\ |S|=m}} \frac{1}{\binom{|N|-1}{|S|}} \left[ \sum_{L \subseteq S} I(L \cup T) \right] \quad \text{// by Lemma 4} \\
=& \frac{k}{|N|} \sum_{L \subseteq N \setminus T} \sum_{m=|L|}^{|N|-k} \frac{1}{\binom{|N|-1}{|S|}} \sum_{\substack{S \subseteq N \setminus T \\ |S|=m \\ S \supseteq L}} I(L \cup T) \\
=& \frac{k}{|N|} \sum_{L \subseteq N \setminus T} \sum_{m=|L|}^{|N|-k} \frac{1}{\binom{|N|-1}{|S|}} \binom{|N|-|L|-k}{m-|L|} I(L \cup T) \\
=& \frac{k}{|N|} \sum_{L \subseteq N \setminus T} I(L \cup T) \underbrace{\sum_{m=0}^{|N|-|L|-k} \frac{1}{\binom{|N|-1}{|L|+m}} \binom{|N|-|L|-k}{m}}_{w_L}
\end{aligned}
$$

Similar to the proof of Theorem 4, we leverage the properties of combinatorial numbers and the Beta function to simplify $w_L$.

$$
\begin{aligned}
w_L =& \sum_{m=0}^{|N|-|L|-k} \frac{1}{\binom{|N|-1}{|L|+m}} \binom{|N|-|L|-k}{m} \\
=& \sum_{m=0}^{|N|-|L|-k} \binom{|N|-|L|-k}{m} \cdot \left( |L|+m \right) \cdot B\left( |N|-|L|-m, |L|+m \right) \\
=& \sum_{m=0}^{|N|-|L|-k} |L| \cdot \binom{|N|-|L|-k}{m} \cdot B\left( |N|-|L|-m, |L|+m \right) \qquad \cdots ① \\
&+ \sum_{m=0}^{|N|-|L|-k} m \cdot \binom{|N|-|L|-k}{m} \cdot B\left( |N|-|L|-m, |L|+m \right) \qquad \cdots ②
\end{aligned}
$$

Then, we solve ① and ② respectively. For ①, we have

$$
\begin{aligned}
① =& \int_0^1 |L| \cdot \sum_{m=0}^{|N|-|L|-k} \binom{|N|-|L|-k}{m} \cdot x^{|N|-|L|-m-1} \cdot (1-x)^{|L|+m-1} \, dx \\
=& \int_0^1 |L| \cdot \underbrace{\left[ \sum_{m=0}^{|N|-|L|-k} \binom{|N|-|L|-k}{m} \cdot x^{|N|-|L|-m-k} \cdot (1-x)^m \right]}_{=1} \cdot x^{k-1} \cdot (1-x)^{|L|-1} \, dx \\
=& \int_0^1 |L| \cdot x^{k-1} \cdot (1-x)^{|L|-1} \, dx = |L| \cdot B(k, |L|) = \frac{1}{\binom{|L|+k-1}{k-1}}
\end{aligned}
$$

For ②, we have

$$\text{②} = \sum_{m=1}^{|N|-|L|-k} (|N|-|L|-k) \cdot \binom{|N|-|L|-k-1}{m-1} \cdot B\Big(|N|-|L|-m, |L|+m\Big)$$

$$= \sum_{m'=0}^{|N|-|L|-k-1} (|N|-|L|-k) \cdot \binom{|N|-|L|-k-1}{m'} \cdot B\Big(|N|-|L|-m'-1, |L|+m'+1\Big)$$

$$= \int_0^1 (|N|-|L|-k) \sum_{m'=0}^{|N|-|L|-k-1} \binom{|N|-|L|-k-1}{m'} \cdot x^{|N|-|L|-m'-2} \cdot (1-x)^{|L|+m'} \, dx$$

$$= \int_0^1 (|N|-|L|-k) \underbrace{\left[ \sum_{m'=0}^{|N|-|L|-k-1} \binom{|N|-|L|-k-1}{m'} \cdot x^{|N|-|L|-m'-k-1} \cdot (1-x)^{m'} \right]}_{=1} \cdot x^{k-1} \cdot (1-x)^{|L|} \, dx$$

$$= \int_0^1 (|N|-|L|-k) \cdot x^{k-1} \cdot (1-x)^{|L|} \, dx = (|N|-|L|-k) \cdot B(k, |L|+1)$$

$$= \frac{|N|-|L|-k}{(|L|+1)\binom{|L|+k}{k-1}}$$

Hence, we have

$$w_L = \text{①} + \text{②} = \frac{1}{\binom{|L|+k-1}{k-1}} + \frac{|N|-|L|-k}{(|L|+1)\binom{|L|+k}{k-1}}$$

$$= \frac{|L|! \cdot (k-1)!}{(|L|+k-1)!} + \frac{|N|-|L|-k}{|L|+1} \cdot \frac{(|L|+1)! \cdot (k-1)!}{(|L|+k)!}$$

$$= \frac{|L|! \cdot (k-1)!}{(|L|+k-1)!} + \frac{|N|-|L|-k}{|L|+k} \cdot \frac{|L|! \cdot (k-1)!}{(|L|+k-1)!}$$

$$= \left[ 1 + \frac{|N|-|L|-k}{|L|+k} \right] \cdot \frac{|L|! \cdot (k-1)!}{(|L|+k-1)!}$$

$$= \frac{|N|}{|L|+k} \cdot \frac{|L|! \cdot (k-1)!}{(|L|+k-1)!}$$

$$= \frac{|N|}{k} \cdot \frac{|L|! \cdot k!}{(|L|+k)!}$$

$$= \frac{|N|}{k} \cdot \frac{1}{\binom{|L|+k}{k}}$$

Therefore, we proved that when $|T| = k$, $I^{\text{Shapley-Taylor}}(T) = \frac{k}{|N|} \sum_{L \subseteq N \setminus T} w_L \cdot I(L \cup T) = \frac{k}{|N|} \sum_{L \subseteq N \setminus T} \frac{|N|}{k} \cdot \frac{1}{\binom{|L|+k}{k}} \cdot I(L \cup T) = \sum_{L \subseteq N \setminus T} \binom{|L|+k}{k}^{-1} I(L \cup T)$.

$\square$

## C  EXPERIMENTAL SETTINGS FOR VISUALIZING INTERACTION PRIMITIVES

### C.1  MODELS AND DATASETS

We used the models (including MLP, ResMLP, LeNet, AlexNet, VGG, PointNet, and PointNet++) provided by Li & Zhang (2023b) and followed their experimental settings. The MLPs and ResMLPs used in this experiment all had 5 fully-connected layers. Each hidden layer had 100 neurons. The *tic-tac-toe* dataset, the *wifi* dataset, and the *phishing* dataset refer to the UCI tic-tac-toe endgame dataset (Dua & Graff, 2017), the UCI wireless indoor localization dataset (Dua & Graff, 2017), and the UCI phishing website prediction dataset (Dua & Graff, 2017), respectively. The *MNIST-3* dataset is a binary classification dataset, where images of the digit "three" in the MNIST dataset (LeCun et al., 1998) were taken as positive samples, while images of other digits were taken as negative samples. The *CelebA-eyeglasses* dataset is a binary classification dataset, where images with the attribute "eyeglasses" in the CelebA dataset (Liu et al., 2015) were taken as positive samples, while

other images were taken as negative samples. The heavy computational cost did not allow us to test on all samples in these datasets. Thus, similar to previous studies Li & Zhang (2023b), we used a subset of samples to compute interactions. For all models except for LLMs, we followed the setting of testing samples in Li & Zhang (2023b), in which 50 samples from the *CelebA-eyeglasses* dataset, 100 samples from the *ShapeNet* dataset and the *MNIST-3* dataset, and 500 samples from each of the three tabular datasets have been selected to compute interactions.

## C.2 THE ANNOTATION OF SEMANTIC PARTS

As mentioned in Section 3.1, given an input sample with $n$ input variables, the DNN may encode at most $2^n$ potential interactions. The computational cost for extracting salient interaction primitives is high, when the number of input variables $n$ is large. For example, if each 3D point of a point-cloud (or each pixel of an image) is taken as a single input variable, the computation is usually prohibitive. In order to overcome this issue, Li & Zhang (2023b) annotated 8-10 semantic parts in each input sample, and the annotated semantic parts are aligned over different samples. Then, each semantic part in an input sample is taken as a "single" input variable to the DNN.

• For the ShapeNet dataset (Yi et al., 2016), Yi et al. have provided annotations for semantic parts for input samples in the *motorbike* category. The semantic parts include *gas tank, seat, handle, light, wheel*, and *frame*. Based on the original annotation, Li & Zhang (2023b) further divided the annotation for each motorbike sample into more fine-grained semantic parts, including *gas tank, seat, handle, light, front wheel, back wheel, front frame, mid frame*, and *back frame*.

• For images in the *MNIST-3* dataset, we used the semantic parts annotated by Li & Zhang (2023b). The annotation of different parts was based on some key points in each image. Figure 6 shows examples of these semantic parts.

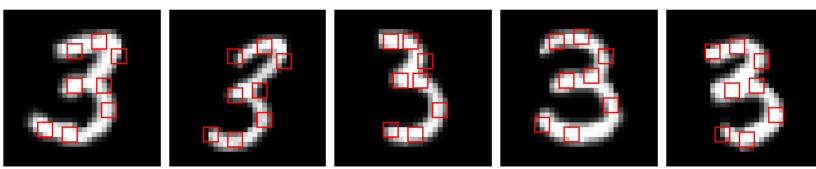

Figure 6: Examples of annotated semantic parts for samples of the *MNIST-3* dataset.

# D  EXPERIMENTAL SETTINGS FOR LARGE LANGUAGE MODELS AND MLPS

## D.1 MODELS AND DATASETS

We used off-the-shelf trained LLMs (including the OPT-1.3B[6] model, the LLaMA-7B[7] model, and BAAI Aquila-7B[8] model) provided by Huggingface in all our experiments. The detailed information on these models and the training data can be found on the corresponding web pages.

To construct input sentences, for each paragraph in the SQuAD dataset, we first took the initial 30 words. Then, starting from the 31st word, we evaluated the following two conditions: 1) the current word had a specific meaning and was not a stop word in NLTK (Bird et al., 2009), 2) there were no punctuations like the period or the semicolon in the five positions preceding the current word. If both conditions were satisfied, we stopped this process and considered the current word as the *target word* for the LLM to predict. We considered the words before this target word as the *input sentence* for the LLM. If either condition was not satisfied, we incorporated the current word into the input sentence and continued evaluating the next word until this process stopped. For each input sentence, we used 10 meaningful words (words that are not NLTK stop words or punctuations) annotated by Shen et al. (2023) to construct the set of input variables $N$, so that we have $n = |N| = 10$. When masking the input sentence, we only masked words in $N$, without changing other "background" words. We tested on the first 1000 sentences in the SQuAD dataset. It was because the heavy computational cost did not allow us to test on all 20000+ sentences in the dataset.

---

[6]`https://huggingface.co/facebook/opt-1.3b`
[7]`https://huggingface.co/linhvu/decapoda-research-llama-7b-hf`
[8]`https://huggingface.co/BAAI/Aquila-7B`

Another potential way to determine the set $N$ is to select regions with significant Shapley values. It is because Theorem 4 shows that the Shapley value $\phi(i)$ can be explained as the result of uniformly assigning attributions of each Harsanyi interaction to each involved variable $i$. Therefore, the Shapley value can serve as a reasonable metric to measure the saliency of input variables.

In order to ensure the inference logic of the network was meaningful, we discarded sentences for which the LLM predicted meaningless words, such as "the", "a", and other stop words. In addition, we discarded sentences on which the classification confidence of the LLM is low. Specifically, we set a threshold $\xi$ for $v(\boldsymbol{x}) - v(\boldsymbol{x}_\emptyset)$. Given an input sentence $\boldsymbol{x}$, if $v(\boldsymbol{x}) - v(\boldsymbol{x}_\emptyset) < \xi$, then we discarded this sentence. We set $\xi = 1$ for the OPT-1.3B model, $\xi = 2$ for the LLaMA-7B model, and $\xi = 3$ for the BAAI Aquila-7B model and the MLP.

### D.2 OR INTERACTIONS AS A SPECIFIC KIND OF AND INTERACTIONS

In addition to the Harsanyi interaction defined in Eq. (1), which represents the AND relationship between input variables, Li & Zhang (2023a) have also defined OR interactions, as follows.

$$I_{\text{or}}(S) = -\sum_{T \subseteq S}(-1)^{|S|-|T|}u(N \setminus T), \quad \text{s.t. } S \neq \emptyset, \tag{61}$$

where $u(N \setminus T) = v(\boldsymbol{x}_{N \setminus T}) - v(\boldsymbol{x}_\emptyset)$. The OR interaction represents the OR relationship between input variables. In particular, $I_{\text{or}}(\emptyset) = u(\emptyset) = 0$. In later discussions, we use $I_{\text{and}}(S)$ to denote the Harsanyi interaction (or the AND interaction) in order to distinguish it from the OR interaction. It is worth noting that an OR interaction can be regarded as a specific AND interaction, if we inverse the definition of the masked state and the unmasked state of an input variable.

**Simultaneously extracting AND-OR interactions.** In fact, a well-trained DNN usually encodes complex interactions between input variables, including both AND interactions and OR interactions. Li & Zhang (2023a) proposed a method to simultaneously extract AND interactions $I_{\text{and}}(S)$ and OR interactions $I_{\text{or}}(S)$ from the network output. Given a set $S \subseteq N$ and the corresponding masked sample $\boldsymbol{x}_S$, Li & Zhang (2023a) proposed to learn a decomposition $u(S) = u_{\text{and}}(S) + u_{\text{or}}(S)$, towards the sparsest interactions. The $u_{\text{and}}(S)$ term was explained by AND interactions, and the $u_{\text{or}}(S)$ term was explained by OR interactions, subject to $I_{\text{and}}(\emptyset) = u_{\text{and}}(\emptyset) = 0, I_{\text{or}}(\emptyset) = u_{\text{or}}(\emptyset) = 0$.

$$u(S) = u_{\text{and}}(S) + u_{\text{or}}(S), \ u_{\text{and}}(S) = \sum_{T \subseteq S} I_{\text{and}}(T), \ u_{\text{or}}(S) = \sum_{T \cap S \neq \emptyset} I_{\text{or}}(T). \tag{62}$$

Specifically, they decomposed $u(S)$ into $u_{\text{and}}(S) = 0.5 \cdot u(S) + \gamma_S$ and $u_{\text{and}}(S) = 0.5 \cdot u(S) - \gamma_S$, where $\{\gamma_S\}_{S \subseteq N}$ are a set of learnable variables that determine the decomposition of output scores for AND and OR interactions. Then, they learned the parameters $\{\gamma_S\}$ by minimizing the following LASSO-like loss to obtain sparse interactions:

$$\min_{\{\gamma_S\}} \sum_{S \subseteq N} |I_{\text{and}}(S)| + |I_{\text{or}}(S)| \tag{63}$$

**Removing noises.** As discussed in Section 3.3, a small noise $\epsilon_S$ in the network output may significantly affect the extracted interactions, especially for high-order interactions. Thus, Li & Zhang (2023a) proposed to learn to remove the noise term $\epsilon_S$ from the computation of AND-OR interactions. Specifically, the decomposition was rewritten as $u_{\text{and}}(S) = 0.5 \cdot (u(S) - \epsilon_S) + \gamma_S$ and $u_{\text{or}}(S) = 0.5 \cdot (u(S) - \epsilon_S) + \gamma_S$. Thus, the parameters $\{\epsilon_S\}$, and $\{\gamma_S\}$ are simultaneously learned by minimizing the loss function in Eq. (63). The values of $\{\epsilon_S\}$ were constrained in $[-\zeta, \zeta]$ where $\zeta = 0.04 \cdot |v(\boldsymbol{x}) - v(\boldsymbol{x}_\emptyset)|$.

When illustrating the near-zero effects of high-order interactions in Figure 4, we extracted both AND and OR interactions (can be regarded as a specific kind of AND interactions) and removed the noises from the network output as mentioned in Section D.2. When computing the number of valid (salient) interactions in Table 1 and 2, we only extracted AND interactions and we still removed the noises from network output.

When comparing the real number of valid interactions with the derived bound in Table 2, we only considered the first $M$-order interactions. This was because the theoretical bound can only be computed for the first $M$-order interactions, due to the Assumption 1-$\alpha$ (interactions higher than the $M$-th order have zero effects). In this way, we only considered the first $M$-order interactions for fair comparison. Nevertheless, it is already shown in Figure 4 that high-order interactions usually had small effects, so ignoring interactions higher than the $M$-th order did not affect the conclusion.

# E  MORE EXPERIMENTAL RESULTS

## E.1  MORE RESULTS TO ILLUSTRATE THE NEAR-ZERO EFFECT OF HIGH-ORDER INTERACTIONS

We visualize the average strength of interactions $I_{\text{str}}^{(m)} = \mathbb{E}_{|S|=m}[|I(S)|]$ of the $m$-th order on more samples. Figure 7, 8, and 9 all show that high-order interactions on these LLMs usually have effects that are close to zero.

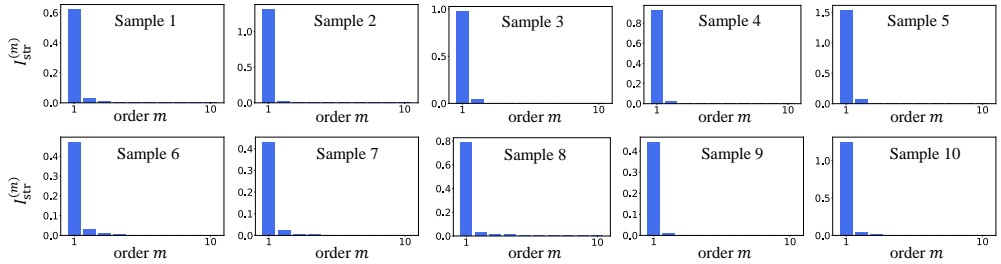

Figure 7: Visualization of the average strength of interactions $I_{\text{str}}^{(m)} = \mathbb{E}_{|S|=m}[|I(S)|]$ of the $m$-th order on the OPT-1.3B model.

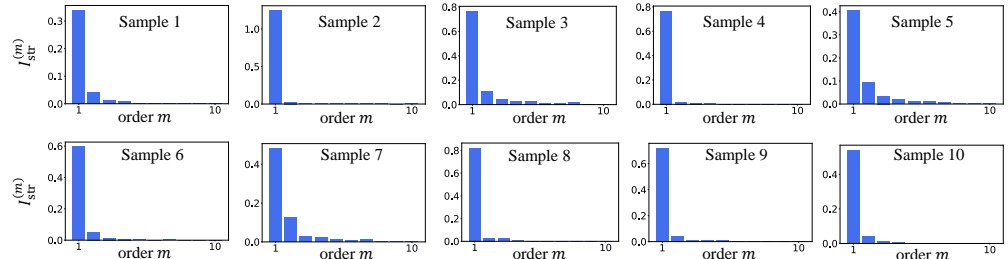

Figure 8: Visualization of the average strength of interactions $I_{\text{str}}^{(m)} = \mathbb{E}_{|S|=m}[|I(S)|]$ of the $m$-th order on the LLaMA-7B model.

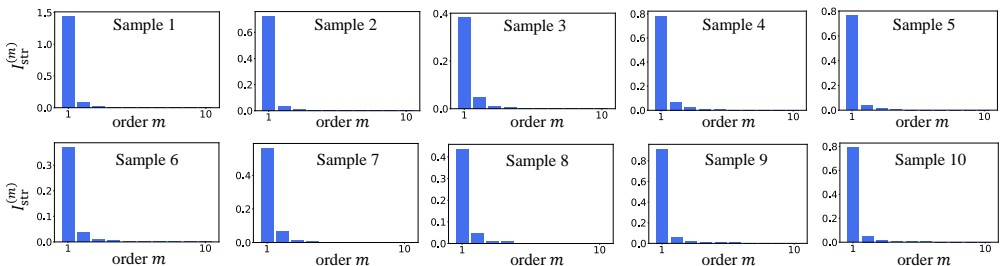

Figure 9: Visualization of the average strength of interactions $I_{\text{str}}^{(m)} = \mathbb{E}_{|S|=m}[|I(S)|]$ of the $m$-th order on the BAAI Aquila-7B model.

## E.2  COMPARISON BETWEEN THE REAL NUMBER OF VALID INTERACTIONS AND THE BOUND ON EXAMPLE SENTENCES

In this section, we show several example input sentences along with their real number of valid (salient) interactions and the derived upper bound. We also show the strength of the normalized interaction $\tilde{I}(S) = I(S)/\max_{S'}|I(S')|$ on these examples in descending order, similar to Figure 3. Figure 10, 11, and 12 shows the results on OPT-1.3B, LLaMA-7B, and BAAI Aquila-7B, respectively.

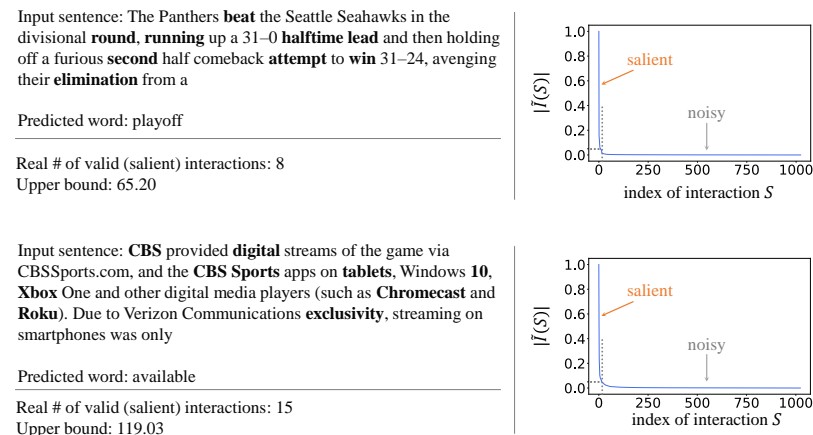

Figure 10: Comparison between the real number of valid interactions and the derived upper bound, and visualization of normalized interaction strength in descending order on the OPT-1.3B model. Words in bold are the input variables in the set $N$ for which we compute interactions.

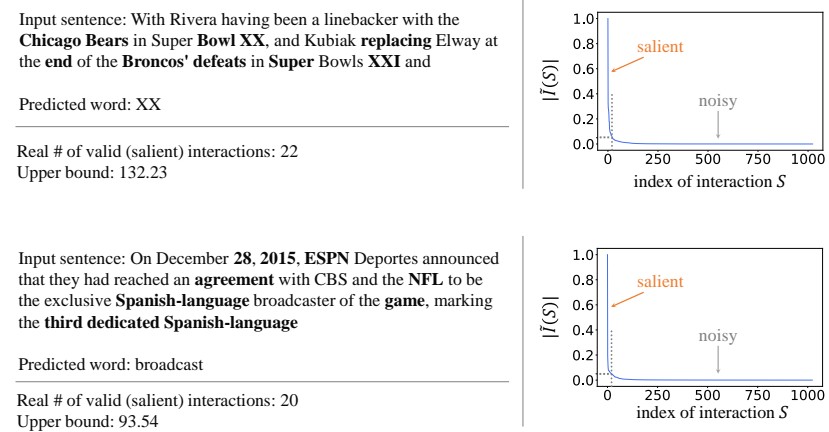

Figure 11: Comparison between the real number of valid interactions and the derived upper bound, and visualization of normalized interaction strength in descending order on the LLaMA-7B model. Words in bold are the input variables in the set $N$ for which we compute interactions.

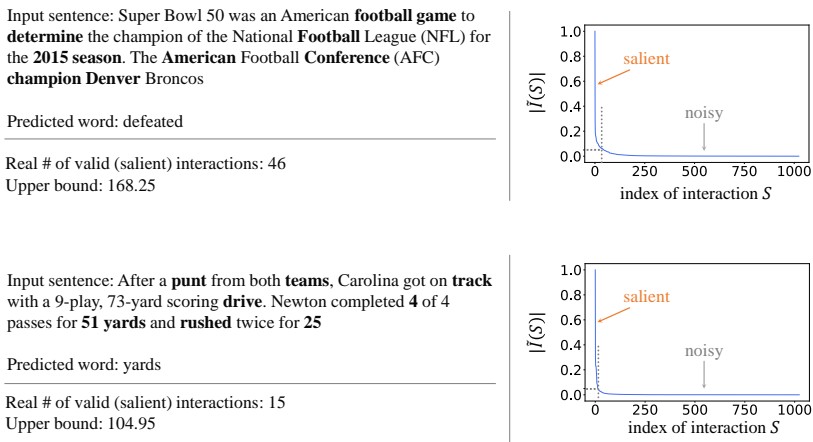

Figure 12: Comparison between the real number of valid interactions and the derived upper bound, and visualization of normalized interaction strength in descending order on the BAAI Aquila-7B model. Words in bold are the input variables in the set $N$ for which we compute interactions.

### E.3 MORE DISCUSSIONS AND EXPERIMENTS ABOUT THE VALUE OF $p$

Although Assumption 3 does not constrain the upper bound for the constant $p$, we have conducted experiments on LLMs to measure the true value of $p$ in real applications. Figure 5(b) shows that the value of $p$ across different samples was around 0.9 to 1.5. According to either Theorem 3 or experimental verification in Table 2, interactions encoded by these LLMs were sparse.

Despite the lack of the bound for $p$, our theory is still complete. It is because the contribution of our study is to clarify the **conditions** that lead to the emergence of sparse interaction primitives, instead of guaranteeing sparse interaction in **every DNN**. In most tasks (*e.g.*, the aforementioned classification tasks), these conditions are commonly satisfied, so that our theory has guaranteed the sparsity of interactions without a need of actually computing the interactions. Section 3.3 discusses a few special cases that do not satisfy the three common conditions. Therefore, instead of showing that sparse interaction primitives will emerge in all scenarios, the goal of this paper is to identify common conditions that provably ensure the emergence of sparse interaction primitives.

In addition, let us analyze how the sparsity of interactions depends on the value of $p$. Some special tasks heavily rely on the global information of all input variables, *e.g.*, the task of judging whether the number of 1's in a binary sequence is odd or even (*i.e.*, judging the parity). Then, in these tasks, the value of $p$ may be large. Therefore, we conducted experiments to train an MLP to judge whether the number of 1's in a binary sequence (*e.g.*, the sequence [0,1,1,1,0,0,1,0,1,1]) is odd or even. Specifically, each binary sequence contains 10 digits. The MLP has 3 layers, and each layer contains 100 neurons. We trained the MLP on 1000 randomly generated samples for 50000 iterations, with the learning rate set to 0.01, and the MLP achieved 100% accuracy on these samples. We then tested the value of $p$ and found that $p$ was around 9.9 to 19.7, which is relatively large.

Fortunately, in most classification tasks as shown in Figure 5(b), the network usually shows a certain level of robustness to the masking of input samples. Therefore, the value of $p$ will not be very large in most classification tasks, thus ensuring the sparsity of interactions encoded by the network.

### E.4 SIZE OF IMAGE REGIONS DOES NOT AFFECT THE SPARSITY OF INTERACTIONS

We would like to clarify that the objective, *i.e.*, the proof of the emergence of sparse interactions, is agnostic to the selection of input variables for interactions. To better illustrate this, we conducted experiments on the *MNIST-3* dataset to show that the interactions were still sparse when the size of image regions varied. Specifically, the original image parts annotated by Li & Zhang (2023b) were of the size of $3 \times 3$ pixels. Then, we enlarged the size of image parts to $5 \times 5$ pixels and $7 \times 7$ pixels. Figure 13 shows that the use of input variables of different sizes did not clearly affect the sparsity of interactions.

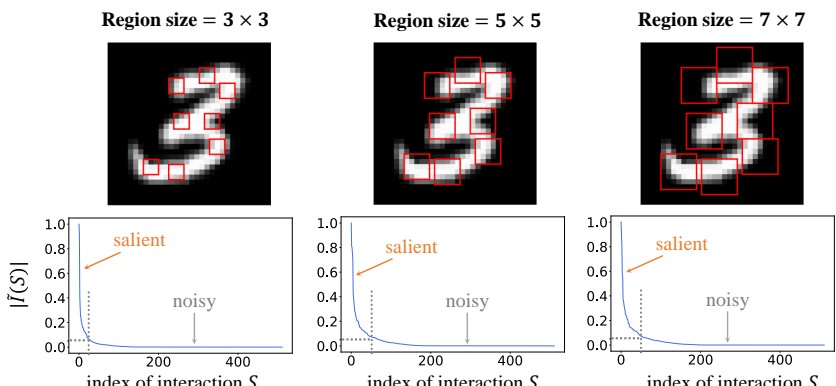

Figure 13: Normalized interaction strength in descending order. We compared the sparsity of interactions when we extracted interactions by setting different sizes of image regions as input variables. Interactions were still sparse when the size of image regions varied.

### E.5    EXPERIMENTS ON RNNS

We conducted experiments on an RNN (*i.e.*, the LSTM) to explore the possibility of applying the Harsanyi interaction to a wider range of tasks. Specifically, we trained LSTMs with 2 layers on the SST-2 dataset (Socher et al., 2013) (for sentiment classification) and the CoLA dataset (Warstadt et al., 2019) (for linguistic acceptability classification). The LSTM can be considered to take natural language as sequential data. Although these tasks are not equivalent to other prediction tasks on sequential data, they provide potential insights into how we can extend the Harsanyi interaction. Figure 14 shows that the network encodes relatively sparse interactions.

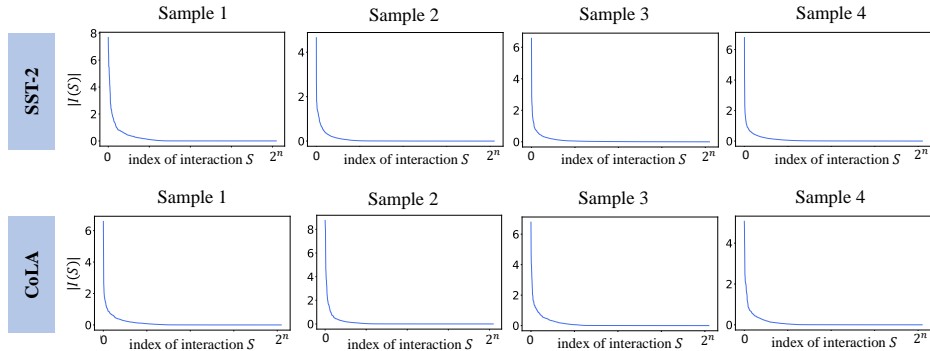

Figure 14: Interaction strength in descending order. The LSTMs also encode sparse interactions.

## F    DISCUSSION ON THE SETTING OF THRESHOLD $\tau$ IN SECTION 3.2

In fact, the setting of $\tau$ is quite reasonable, because most interactions with $|I(S)| < \tau$ actually have zero interaction effect $I(S) = 0$, instead of having an extremely small yet non-zero effect by coincidence. According to $I(S) = \sum_{T \subseteq S} (-1)^{|S|-|T|} \cdot u(T)$, interaction $I(S)$ is computed by adding and subtracting an exponential number of non-zero terms. In this case, we usually obtain two types of interaction effects. First, we may obtain $I(S) = 0$, if the DNN does not encode an AND relationship between an exact set of variables in $S$. Then, different $u(T)$ values for $T \subseteq S$ will eliminate each other, as long as the DNN does not bring random noises to the output $u(T)$. Alternatively, we may obtain $I(S)$ with considerable value $|I(S)| \geq \tau$. In real applications, it is unrealistic for a DNN to be so elaborately trained that adding up $2^{|S|}$ terms results in an extremely small yet non-zero interaction $I(S)$, although we do not fully deny the negligible possibility that we may have a few interactions with extremely small interaction effects.

## G  BROADER IMPACT OF PROVING THE EMERGENCE OF SYMBOLIC (SPARSE) INTERACTIONS

Proving the emergence of symbolic (sparse) interactions provides an alternative methodology for deep learning, *i.e.*, *communicative learning*. In contrast to end-to-end learning from the data, in communicative learning, people may directly communicate with the middle-level concepts encoded by the DNN to examine and fix the representation flaws of the DNN. Communicative learning may include, but is not limited to, (i) the extraction and visualization of symbolic concepts, (ii) the alignment of such implicitly encoded concepts and explicitly annotated human knowledge, (iii) the diagnosis of representation flaws of a DNN, (iv) the discovery of new concepts from DNNs to enrich human knowledge, and (v) interactively fixing/debugging incorrect concepts in a DNN.

## H  AN EXAMPLE TO ILLUSTRATE THE VALIDITY OF ASSUMPTION 2

Assumption 2 assumes that the average network output $\bar{u}^{(m)} = \mathbb{E}_{|S|=m}[u(S)]$ monotonically increase with the order $m$, which considers all $\binom{n}{m}$ possible subsets $S$ with size $m$, and the average network output is much stabler and more robust than the output on a specific masked state $u(S)$. *This assumption is common for most models, without requiring all input variables to carry useful information.* For example, let us explain the target model $v(x) = x_1 x_2 x_3 + x_1 x_2 + x_2 x_3 + x_2 + x_3$, where the input $x = [x_1, x_2, x_3, x_4, x_5]$ contains 5 input variables indexed by $N = \{1, 2, 3, 4, 5\}$, and each input variable $x_i \in \{0, 1\}$ is binary. Here, $x_1$, $x_2$, and $x_3$ are related to the classification, while $x_4$ and $x_5$ are unrelated to the classification. Then, for $|S| = 2$, the value of all possible $u(S)$ are listed as follows: $u(\{1, 2\}) = 2$, $u(\{1, 3\}) = 1$, $u(\{1, 4\}) = 0$, $u(\{1, 5\}) = 0$, $u(\{2, 3\}) = 3$, $u(\{2, 4\}) = 1$, $u(\{2, 5\}) = 1$, $u(\{3, 4\}) = 1$, $u(\{3, 5\}) = 1$, $u(\{4, 5\}) = 0$. Similarly, for $|S| = 3$, the value of all possible $u(S)$ are listed as follows: $u(\{1, 2, 3\}) = 5$, $u(\{1, 2, 4\}) = 2$, $u(\{1, 2, 5\}) = 2$, $u(\{1, 3, 4\}) = 1$, $u(\{1, 3, 5\}) = 1$, $u(\{1, 4, 5\}) = 0$, $u(\{2, 3, 4\}) = 3$, $u(\{2, 3, 5\}) = 3$, $u(\{2, 4, 5\}) = 1$, $u(\{3, 4, 5\}) = 0$. We can see that $\mathbb{E}_{|S|=2}[u(S)] = 1 \leq 1.8 = \mathbb{E}_{|S|=3}[u(S)]$, which satisfies the monotonicity assumption.

## I  CAN WE HAVE AN EFFICIENT WAY TO VERIFY ASSUMPTION 2 AND ASSUMPTION 3 (APPROXIMATING $p$ VALUE) ON A SPECIFIC SAMPLE

In order to efficiently test whether Assumption 2 and Assumption 3 are satisfied, given an input sample $x$, we can simply sample a set of subsets (not all subsets) $\{S_1, S_2, \cdots, S_t\}$ for each order $m$ *s.t.* $|S_i| = m$, to approximate $\bar{u}^{(m)}$ and the value of $p$, which significantly reduces the computational cost. Mathematically, we have $\bar{u}^{(m)} \approx \frac{1}{t} \sum_{i=1}^{t} u(S_i)$, where $|S_i| = m$ for $1 \leq i \leq t$. Then, we can simply use the above approximated $\bar{u}^{(m)}$ to compute the value $p$. In this way, the computational cost of empirically testing the monotonic increase of $\bar{u}^{(m)}$ and a rough estimation of the value of $p$ is $\mathcal{O}(nt)$, which is much less than $\mathcal{O}(2^n)$.

