# OpenReview forum: "Where We Have Arrived in Proving the Emergence of Sparse Interaction Primitives in DNNs"
_ICLR.cc/2024/Conference — ICLR 2024 poster_

### Official Review · Reviewer_iVY1 · 2023-10-28

**Soundness:** 2 fair
**Presentation:** 3 good
**Contribution:** 2 fair
**Rating:** 6
**Confidence:** 2

**Summary:**

This paper attempts to prove the emergence of sparse interaction primitive inference patterns in well-trained AI models. Core to this proof, is three necessary conditions that must be satisfied for this sparse interaction between input variables.

**Strengths:**

* The paper is well-written and clear.
* The paper contributes novel proofs for theorems 2 onwards.
* The paper appears well-placed in the relevant literature.
* The papers theoretical results appear significant and relevant to the ICLR community.

**Weaknesses:**

* The claim of to "prove the emergence of symbolic concepts (or more precisely, sparse primitive inference patterns) in well-trained AI models." (Line 1), appears too broad and un-substantiated, specifically encompassing "AI models". It appears the evidence presented in the paper is only applicable and demonstrated for Deep Neural Networks (DNNs). For example, I cannot see how this proof or claims can be applied to other AI models that are not DNNs, such as Random Forests, which considers the outputs of all trees; k-NN's, that use all features to compute distances, or Boosting algorithms (e.g., Gradient Boosting Machines). Specifically, the empirical evidence presented in the paper in Figure 3, Figure 4, Figure 5 and Table 1, only apply to deep neural network models, and there is no other AI models that are not DNN's to support this claim.
* It is not clear how the input variables should be grouped to form the set of $S$ input variables. For example, in the dog image example, "image regions" are proposed; however, there is no indication or discussion on how large these "image regions" should be for the approach and proof to work. For example, does this sparse primitive approach still hold if the "image regions" are too small, e.g., each region is a single pixel, and does it still hold if the "image regions" are too large, e.g., encompassing most of the image, or even half of the image. Some discussion on this, and how to select these groupings could be informative to the reader.
*  Missing error bars; and large error bars. There appears to be missing error bars for Figure 4, Figure 5 (a), Table 1 (top row: Percent of samples with monotonicity). The provided error bars in Table 1 and Table 2, are so large that they overlap; leading the reader to suspect that the claim of the input samples satisfying the monotonicity assumption is not empirically verified, as the numeric results in Table 1 are not statistically significant, as the error bars overlap.

**Questions:**

* Can you clarify if the core proof of the paper applies to other AI models that are not DNN's? If it does, can you provide evidence for this, and if it does not, perhaps think of refining your claims, and title to be only applicable to DNNs.
* Can you clarify how the reader should group input variables to form the set of $S$?
* Can you include error bars for the top row of Table 1, and include error bars for Figure 4 and Figure 5 (a). Also can you re-run with more random seeds to see if the error bars in Table 1 reduce.

---

> ### Author Response · Authors · 2023-11-17
> **Response to Reviewer iVY1 (Part 1)**
>
> Thank you for your great efforts on the review and constructive comments. We will try our best to answer all your questions.
>
> **If you are not satisfied with our answers or have more questions, please let us know as soon as possible, so that we can try our best to answer any further questions before the deadline.**
>
> $\textcolor{blue}{\textsf{Q1: Ask about the use of the phrase ``AI models" in our title and claims.}}$
>
> > "The claim of to "prove the emergence of symbolic concepts (or more precisely, sparse primitive inference patterns) in well-trained AI models." (Line 1), appears too broad and un-substantiated, specifically encompassing 'AI models'.  "
> >
> > "Can you clarify if the core proof of the paper applies to other AI models that are not DNN's? If it does, can you provide evidence for this, and if it does not, perhaps think of refining your claims, and title to be only applicable to DNNs.   "
>
> A: Thank you. We have followed your suggestions to revise the phrase "AI model" in all claims to the phrase "DNN" in the paper. Please see the revised paper.
>
> ---
>
> $\textcolor{blue}{\textsf{Q2: Ask about the selection of input variables in images.}}$
>
> > "It is not clear how the input variables should be grouped to form the set $S$ of input variables."
> >
> > "For example, does this sparse primitive approach still hold if the 'image regions' are too small, e.g., each region is a single pixel, and does it still hold if the 'image regions' are too large, e.g., encompassing most of the image, or even half of the image. "
> >
> > "Can you clarify how the reader should group input variables to form the set of $S$?  "
>
> A: Thank you for your comments.  Thank you for your comments. Before the detailed answer, we need to clarify that the proof of our theory does not depend on the definition of input variables. That is, no matter how we select input variables in the input, the DNN usually encodes sparse interactions, as long as it satisfies the proposed three common conditions.
>
> Then, we are glad to introduce the detailed settings. For images in the MNIST-3 dataset, we used the semantic parts that had been annotated by [cite 1]. The annotation of different parts was based on some keypoints in each image. We have revised the paper to show annotated parts in some MNIST images. Please see Appendix C.2 and Figure 6 in the updated PDF for details.
>
> **New experimental verification.** In addition, we have also conducted **new experiments** on the MNIST-3 dataset to show that the interactions were still sparse when the size of image regions varies. Specifically, original image parts annotated by [cite 1] were of the size of $3\times 3$ pixels. Then, we enlarged the size of image parts to $5\times 5$ pixels, and $7\times 7$ pixels. Figure 13 in Appendix E.4 in the update PDF shows the normalized interaction strength in descending order under different settings of the size of image regions. Under image regions (input variables) of different sizes, we always extracted sparse interactions.
>
> [cite 1] Mingjie Li and Quanshi Zhang. Does a Neural Network Really Encode Symbolic Concept? ICML, 2023.

---

> ### Author Response · Authors · 2023-11-17
> **Response to Reviewer iVY1 (Part 2)**
>
> $\textcolor{blue}{\textsf{Q3: Ask about error bars.}}$
>
> > "Missing error bars; and large error bars. There appears to be missing error bars for Figure 4, Figure 5 (a), Table 1 (top row: Percent of samples with monotonicity). The provided error bars in Table 1 and Table 2, are so large that they overlap,  leading the reader to suspect that the claim of the input samples satisfying the monotonicity assumption is not empirically verified, as the numeric results in Table 1 are not statistically significant.  "
> >
> > "Can you include error bars for the top row of Table 1, and include error bars for Figure 4 and Figure 5 (a). Also can you re-run with more random seeds to see if the error bars in Table 1 reduce. "
>
> A: Thank you. We have followed your suggestions to add error bars in Figure 4 and Figure 5(a). For Figure 4, we plot the Box-and-whisker diagram for the strength of interactions $|I(S)|$ of each order $m$, which roughly demonstrates the distribution of the strengh of interactions of different orders. For Figure 5(a), the shaded area measures the standard deviation of all $m$-order network outputs on a specific sample, i.e., $Std_{|S|=m}[u(S)]$, where $u(S)=v(x\_S)-v(x\_\emptyset)$ as mentioned in the paragraph below Equation (1). For the top row in Table 1, there does not exist an error bar, because the first row reports the ratio of samples in the dataset that satisfy the monotonicity assumption, which is a deterministic number without variances.
>
> For your concerns on the other rows in Table 1, we have revised the organization of Table 1 to answer your question. The second row reports the average number of valid interactions, when the sample satisfied the monotonicity. The third row reports the number of average number of valid interactions, when the sample did not satisfy the monotonicity. It is meaningless to compare numbers in the 2nd and 3rd rows. Instead, **the two rows show that we could obtain sparse interactions even on samples that did not  satisfy the monotonicity.** It is because the monotonicity assumption is just a sufficient condition for the sparsity of interactions, rather than a necessary condition. Therefore, Table 1 actually shows that both samples with monotonicity and samples without monotonicity all usually exhibited sparse interactions. The comparison of the average number of valid interactions between the 2nd and 3rd rows cannot either validate or invalidate the monotonicity assumption. **Instead, we use the high ratio of samples with monotonicity (1st row of Table 1, 85.33% -- 90.26%) to prove the universality of the monotonicity assumption.**
>
>
>
> **If our understanding of your questions is incorrect, please give us feedback as soon as possible. We will try our best to answer any further questions.**

---

> > ### Comment · Reviewer_iVY1 · 2023-11-22
> > **Reviewer iVY1 Response**
> >
> > Thank you for your detailed comments; they have now addressed my concerns. As such, I have raised my score to reflect this. Thank you for including the error bars and explaining my misunderstanding with the error bars. I believe these changes and the refined claim of only applying to DNNs improve this work.

---

### Official Review · Reviewer_JPCH · 2023-10-30

**Soundness:** 4 excellent
**Presentation:** 4 excellent
**Contribution:** 4 excellent
**Rating:** 8
**Confidence:** 3

**Summary:**

The authors set out to prove that the knowledge in a deep neural network can be understood as a function of very few sparse interactions as compared to the total possible large space of possible interactions between input features. They define these interactions through the Harsanyi dividend, and make assumptions under which they prove that the total number of salient interactions is upper bounded by a quantity which is much smaller than the total possible interactions. These assumptions are tested empirically, and the proof shows that neural networks learn to represent complex functions using few symbolic rules.

**Strengths:**

1) I found the paper very interesting to read. The paper is well motivated and the authors tackle the important but challenging problem of trying to show that deep neural networks learn symbolic concepts, and not just general fuzzy and arbitrarily complex interaction functions of their inputs.

2) I found the assumptions made by the authors in the proof to be generally reasonable, and the authors give satisfactory empirical evidence that these assumptions hold for some important classes of networks.

3) The proof was worked through quite clearly, and I found I could follow along reasonably well without needing to go through related works despite not having prior background knowledge in this subfield.

**Weaknesses:**

While the work shows that the number of valid interactions tends to be much smaller than the total possible set of interactions, this does not mean that the number of interactions is small in an absolute sense. For high dimensional inputs such as images, there can still be an extremely large number of interactions depending on the level of abstraction of the concepts being looked at to describe these interactions. For images, would pixels, patches, or some general localized features be the right abstraction to view interactions? As such, I am not sure if thinking of neural networks as learning these sparse interactions is a useful perspective.

**Questions:**

1) I would like the authors to comment on the weakness I mentioned above.

While I am not very familiar with the background work in this field, the paper tackles an important problem in deep learning, and does a convincing job justifying its claims. As such I am recommending acceptance.

---

> ### Author Response · Authors · 2023-11-20
> **Response to Reviewer JPCH (Part 1)**
>
> Thank you very much. We have found that many comments have deep insights. We are glad to answer all your questions.
>
> **If you are not satisfied with our answers or have more questions, please let us know as soon as possible, so that we can try our best to answer any further questions before the deadline.**
>
> $\textcolor{blue}{\textsf{Q1:}}$
>
> > " While the work shows that the number of valid interactions tends to be much smaller than the total possible set of interactions, this does not mean that the number of interactions is small in an absolute sense. For high-dimensional inputs such as images, there can still be an extremely large number of interactions depending on the level of abstraction of the concepts being looked at to describe these interactions. For images, would pixels, patches, or some general localized features be the right abstraction to view interactions? As such, I am not sure if thinking of neural networks as learning these sparse interactions is a useful perspective. "
>
> A: Yes, your understanding is correct. Although we have proven the sparsity of interactions, the total number of all interactions still depend on how we define the basic input variables.
>
> $\bullet$ **Experimental examination of the sparsity of interactions when we set different input variables.** We have conducted **a new experiment** on the MNIST dataset, in which we tried to use image patches of different sizes as input variables.  Specifically, original image patches annotated by Li & Zhang (2023) were of the size of $3\times 3$ pixels. Then, we enlarged the size of image patches to $5\times 5$ pixels, and $7\times 7$ pixels. Figure 13 in Appendix E.4 in the revised paper compares the normalized interaction strength in descending order, when interactions were extracted under different settings of the size of image patches. It indicates that given image patches of different sizes, we still extracted interactions of similar sparsity. Please see Appendix E.4 for details.
>
> $\bullet$ **How to set input variables in real applications.** The exponential computational cost makes us unable to compute the exact interactions when there are a large number of input variables. However, in real applications, we do not need to take all input dimensions as input variables, so that we usually do not obtain a large number of input variables. We can have different ways to annotate input variables and reduce the number of input variables. Please see Appendix C.2 and the second and third paragraphs of Appendix D.1 for details. In addition, we can only set important input dimensions as input variables and take other input dimensions unrelated to the task as constant background (never be masked in $v(x_T)$). A typical way is to select input dimensions with significant Shapley values. It is because Theorem 4 shows $\phi(i)=\sum_{S\subseteq N\setminus \\{i\\}}\frac{1}{|S|+1} I(S\cup \\{i\\})$, which means that the Shapley value $\phi(i)$ can be explained as the result of uniformly assigning each Harsanyi interaction $I(S\cup\\{i\\})$ to each involved variable. Therefore, the Shapley value $\phi(i)$ can serve as a reasonable metric to measure the saliency of input variables. We can simply select input dimensions with considerable Shapley values as input variables. We have also added the way of using Shapley values for variable selection in the third paragraph of Appendix D.1 in the revised paper.
>
> $\bullet$ **Significance of proving sparse interactions.** In fact, the focus of our study, i.e., whether the inference logic of a DNN can be faithfully represented as symbolic primitives, is one of the fundamental problems in the field of explainable AI. Many previous studies have attempted to explain the feature representation of DNNs by using symbolic concepts with clear meanings. For example, Bau et al. (2017) tried to visualize the top-activated receptive fields corresponding to each convolutional filter to represent the concept encoded by the filter. Kim et al. (2018) assumed that there was a certain feature direction in the intermediate layer corresponding to each manually defined concept. These methods all potentially assumed that DNNs indeed encoded a small number of symbolic concepts rather than a massive number of fuzzy features. **Therefore, if the emergence of sparse concepts is disproved, then it will lose theoretical support to faithfully explain DNN representations into semantic concepts**, although many studies have not formally discussed the faithfulness of the explanation. To this end, although it is hard to say our paper is the ultimate proof of the emergence of symbolic primitives in a DNN, our study proves that under three common conditions, a DNN encodes sparse symbolic interaction primitives. This can partially serve as some support for all attempts to explain semantics encoded by a DNN.

---

> ### Author Response · Authors · 2023-11-20
> **Response to Reviewer JPCH (Part 2)**
>
> (continued for Q1)
>
> $\bullet$ **Using interactions to explain the generalization power of a DNN.** In addition, some people have used interactions to redefine the generalization power of a DNN (Zhou et al., 2023). Unlike traditional studies on the generalization power of a DNN (e.g., focusing on the gap between training and testing losses), this line of research is led by a different motivation, i.e., clarifying and exploring the detailed inference logics encoded by a DNN, which are responsible for its generalization power. To this end, despite the huge gap between symbolism and connectionism, the proof of the emergence of the sparsity of interactions in this study provides a new potential for the challenging task. Specifically, given multiple DNNs trained for the same task and an input sample, if an interaction can be extracted from all/most of these DNNs, then we consider it generalizable. In this way, we can identify a set of generalizable interactions and a set of non-generalizable interactions, which show much deeper insights into the generalization power of a DNN. To this end, the proven sparsity of interactions may further boost the trustworthiness of using interactions to explain the generalization power of a DNN.
>
> In addition, ensuring the interactions' generalization power may serve as the last piece to prove the faithfulness of using symbolic interactions to explain a DNN. We may consider that we have faithfully defined interaction primitives encoded by a DNN, if we simultaneously ensure that (1) each DNN just encodes a small number of salient interactions (partially solved in this paper), (2) the small number of salient interactions can universally match the DNN outputs on all randomly masked samples (proven by Li & Zhang (2023)), (3) most salient interactions are generalizable over different DNNs  trained on the same task (discovered by Li & Zhang (2023)).
>
> (Li & Zhang, 2023) Mingjie Li and Quanshi Zhang. Does a Neural Network Really Encode Symbolic Concept? ICML, 2023.
>
> (Bau et al., 2017) David Bau, Bolei Zhou, Aditya Khosla, Aude Oliva, and Antonio Torralba. Network Dissection: Quantifying Interpretability of Deep Visual Representations. CVPR, 2017.
>
> (Kim et al., 2018) Been Kim, Martin Wattenberg, Justin Gilmer, Carrie Cai, James Wexler, Fernanda Viegas, and Rory Sayres. Interpretability Beyond Feature Attribution: Quantitative Testing with Concept Activation Vectors (TCAV). ICML, 2018.
>
> (Zhou et al., 2023) Huilin Zhou, Hao Zhang, Huiqi Deng, Dongrui Liu, When Shen, Shih-Han Chan, and Quanshi Zhang. Concept-Level Explanation for the Generalization of a DNN. arXiv preprint arXiv:2302.13091, 2023.

---

> > ### Comment · Reviewer_JPCH · 2023-11-21
> >
> > I thank the authors for their detailed response. I will keep my score.

---

### Official Review · Reviewer_8DUx · 2023-10-31

**Soundness:** 3 good
**Presentation:** 3 good
**Contribution:** 2 fair
**Rating:** 6
**Confidence:** 3

**Summary:**

The paper under review studied the emergence of symbolic concepts of deep neural networks.
Consider the set of subsets of each input data (e.g. an image or a sentence), the authors defined interaction between these subsets.
Then, the authors zoomed into DNN models satisfied three proposed conditions, showed that sparse positive interactions emerge under these conditions.
In addition, practicality of the proposed conditions are illustrated by simulations.

**Strengths:**

The idea of qualitatively measure whether a DNN model can emerge a few symbolic concepts is novel and grounded.
The paper is well written, easy to follow. Plots are very illustrative and helpful.

**Weaknesses:**

1. Soundness of assumption 2 and 3:

- assumption 2 assumes the more unmasked subsets of an input, the higher expected output. Intuitively, if each subset carries useful info, then yes.
However, in many situation that is not the case. For instance, even in Figure 2, three unmasked regions (row 2) yield a much smaller output than two unmasked regions (row 1). The authors also mention that if there are many noisy regions, then only focus on the salient regions.
In practice, how to does one know such info apriori?

- assumption 3 assumes there exists a constant $p$ such an inequality hold. But no bound on $p$ is derived.
From the definition, $p$ can be arbitrarily large. Then it is not clear how Thm 2 shows the sparsity of interactions for models satisfies assumption 1-3.
(See more details in Question section.)

2. These lead to my major concern: practicality. While intuitively, the proposed three criterion all make sense, however, for a general DNN model, how to validate these three conditions, besides empirically testing after training, is somehow unclear.


3. Implication of the work: my understanding is that the authors aim to derive conditions, under which DNN models will produce sparse positive interactions (considered as small number of concepts). As detailed above, if one can't test these conditions prior training, how would these conditions be used? On the other hand, it would be nice if the authors could comment on implications of Thm 3, i.e. how would knowing many DNN are able to learn a few key concepts benefit us?

**Questions:**

1. Figure 3: would the authors please provide more experiment details? Didn't find much in appendix neither. Here are a few main questions:
 - did the authors train the 5 mentioned models or existing trained model were used?
 - Size of training, testing data.
 - For each of the input, how was the subsets selected? for instance, for a image from MNIST, following the paper notation $\mathbf{x} = (x_1, \dots, x_n)$, is each $x_i$ just a pixel?


2. Thm 2 and Thm 3: any insight on coefficients $\lambda$'s in eq(4) and eq (7)? How Thm2 and Thm 3 imply sparsity are unclear.
Based on eq (7), valid number of k-th order interactions is bound by roughly a constant times $\lambda^{(k)} * n^{p+\delta}$.
On one hand, $\lambda^{(k)}$ can be large. On the other hand, the authors claims $n^{p+\delta}$ is much less than
$n \choose k$. Yet bound on $p$ is not required, in case where say $n-k < p$, not sure how the claim holds.

---

> ### Author Response · Authors · 2023-11-17
> **Response to Reviewer 8DUx (Part 1)**
>
> Thank you for your great efforts on the review and constructive comments. We will try our best to answer all your questions.
>
> **If you are not satisfied with our answers or have more questions, please let us know as soon as possible, so that we can try our best to answer any further questions before the deadline.**
>
> $\textcolor{blue}{\textsf{Q1: Ask about Assumption 2.}}$
>
> > "Assumption 2 assumes the more unmasked subsets of an input, the higher expected output. Intuitively, if each subset carries useful info, then yes. However, in many situation that is not the case. For instance, even in Figure 2, three unmasked regions (row 2) yield a much smaller output than two unmasked regions (row 1). The authors also mention that if there are many noisy regions, then only focus on the salient regions. In practice, how to does one know such info apriori? "
>
> A: Thank you for your comments. Maybe our presentation is not clear enough, and it seems our presentation leads to a misunderstanding of Assumption 2 and Figure 2.
>
> $\bullet$ First, let us answer your concern about the validity of Assumption 2, which is also discussed in Appendix H in the updated PDF of the paper. Assumption 2 assumes that the average model output $\bar{u}^{(m)}=\mathbb{E}\_{|S|=m}[u(S)]$ monotonically increase with the order $m$, which considers all $\binom{n}{m}$ possible subsets $S$ with size $m$, and the average network output is much stabler and more robust than the output on a specific masked state $u(S)$. *This assumption is common for most models, without requiring all input variables to carry useful information.* For example, let us explain the target model $v(x)=x_1 x_2 x_3 + x_1 x_2 + x_2 x_3 + x_2 + x_3$, where the input $x=[x_1,x_2,x_3,x_4,x_5]$ contains 5 input variables indexed by $N=\\{1,2,3,4,5\\}$, and each input variable $x_i\in\\{0,1\\}$ is binary. Here, $x_1$, $x_2$, and $x_3$ are related to the classification, while $x_4$ and $x_5$ are unrelated to the classification. Then, for $|S|=2$, the value of all possible $u(S)$ are listed as follows: $u(\\{1,2\\})=2$, $u(\\{1,3\\})=1$, $u(\\{1,4\\})=0$, $u(\\{1,5\\})=0$, $u(\\{2,3\\})=3$, $u(\\{2,4\\})=1$, $u(\\{2,5\\})=1$, $u(\\{3,4\\})=1$, $u(\\{3,5\\})=1$, $u(\\{4,5\\})=0$. Similarly, for $|S|=3$, the value of all possible $u(S)$ are listed as follows: $u(\\{1,2,3\\})=5$, $u(\\{1,2,4\\})=2$, $u(\\{1,2,5\\})=2$, $u(\\{1,3,4\\})=1$, $u(\\{1,3,5\\})=1$, $u(\\{1,4,5\\})=0$, $u(\\{2,3,4\\})=3$, $u(\\{2,3,5\\})=3$, $u(\\{2,4,5\\})=1$, $u(\\{3,4,5\\})=0$. We can see that $\mathbb{E}\_{|S|=2}[u(S)]=1 \le 1.8= \mathbb{E}\_{|S|=3}[u(S)]$, which satisfies the monotonicity assumption. This is just a simple example. In fact, we do not fully understand your mentioned case for the failure of Assumption 2. Please let us know and provide more details if we misunderstand your concerns.
>
> We have also carefully revised the presentation to clarify the problem (please see the second paragraph below Assumption 2 in the updated PDF for details).
>
> $\bullet$ Second, let us answer your concern on Figure 2. The mentioned phenomenon that *three unmasked regions (row 2) yield a much smaller "output" than two unmasked regions (row 1)* in Figure 2 does **not** conflict with Assumption 2. It is because the mentioned "output" in Figure 2 is actually the interaction effect $I(S)$ of a specific subset $S$, rather than the network output score $u(S)$. Assumption 2 claims the *monotonical increase of the average model output* $\mathbb{E}\_{|S|=2}[u(S)]\le \mathbb{E}\_{|S|=3}[u(S)]$, rather than the *monotonical increase of interaction effects* $\mathbb{E}\_{|S|=2}[I(S)]\le \mathbb{E}\_{|S|=3}[I(S)]$. Thus, Figure 2 shows that effects of 3-order interactions $I(S)$ were smaller than those of 2-order interactions, which did not conflict with Assumption 2.
>
> $\bullet$ Third, let us answer the selection of salient regions. In the beginning, let us clarify that the proof of the sparsity of interactions is agnostic to the selection of salient regions. However, when we conducted experimental verification of our theory, we had to select a set of salient variables (salient regions) to reduce the huge computational cost, although the variable selection was not directly related to the proof of our theory. In fact, we have many different ways to select salient regions from noisy regions. The second paragraph in Appendix D.1 shows how we selected the set $N$ from all words in the input sentence. Specifically, for each input sentence, we used 10 meaningful words (words that are not NLTK stop words or punctuations) annotated by [cite 1] to construct the set of input variables $N$. Please see the second paragraph in Appendix D.1 for these.
>
> [cite 1] Wen Shen, Lei Cheng, Yuxiao Yang, Mingjie Li, and Quanshi Zhang. Can the Inference Logic of Large Language Models be Disentangled into Symbolic Concepts? arXiv preprint arXiv:2304.01083, 2023.

---

> > ### Comment · Reviewer_8DUx · 2023-11-21
> > **Concern regarding assumption 2**
> >
> > Thank you for the detailed response. It cleared my confusion on figure 2.
> > However, it does not fully address my concern on the soundness of the monotonicity assumption (assumption 2).
> >
> > The authors mentioned in the last paragraph on page 6,
> > quote"Note that this monotonicity assumption corresponds to a  __relatively ideal case__. If many variables in the input sample represent the background or are not directly related to the target class, or if the model is not well-trained, then reducing the number of masked variables in the input sample may not necessarily increase the model’s classification confidence....."
> > My concern is along the same line. Since we can't really check this monotonicity prior training. Assumption 2 reads somewhat backward to me: if a model is well-trained, we sort of already know the model has learned something. Then the value of checking proposed assumptions is somewhat less important, as the authors do not have a bound on the number of concepts learned (no bound of p is proved).

---

> ### Author Response · Authors · 2023-11-17
> **Response to Reviewer 8DUx (Part 2)**
>
> (continued for Q1) Besides, another typical way is to select regions with significant Shapley values. It is because Theorem 4 shows that the Shapley value $\phi(i)$ can be explained as the result of uniformly assigning attributions of each Harsanyi interaction to each involving variable $i$. Therefore, the Shapley value can serve as a reasonable metric to measure the saliency of input variables. We have also added the way of using Shapley values for variable selection in the third paragraph of Appendix D.1 in the updated PDF.
>
> ---
>
> $\textcolor{blue}{\textsf{Q2: Ask about the value of $p$}}.$
>
> > "Assumption 3 assumes there exists a constant $p$ such an inequality hold. But no bound on $p$ is derived. From the definition, $p$ can be arbitrarily large."
>
> A: A good question. Although Assumption 3 does not constrain the upper bound for the constant $p$, we **have conducted experiments** on Large Language Models (LLMs) to measure the true value of $p$ in real applications. Figure 5(b) shows that the value of $p$ across different samples was around 0.9 to 1.5. According to either Theorem 3 or experimental verification in Table 2, interactions encoded by these LLMs were sparse.
>
> More importantly, despite the lack of the bound for $p$, our theory is still complete. It is because the contribution of our study is to clarify the **conditions** that lead to the emergence of sparse interaction primitives, instead of guaranteeing sparse interaction in **every DNN**. In most tasks (e.g., the aforementioned classification tasks), these conditions are commonly satisfied, so that our theory has guaranteed the sparsity of interactions without a need to actually compute the interactions. Section 3.3 discusses a few special cases that do not satisfy the three common conditions. Therefore, instead of showing that sparse interaction primitives will emerge in all scenarios, the goal of this paper is to identify common conditions that provably ensure the emergence of sparse interaction primitives.
>
> In addition, let us conduct **a new experiment** to help explain how the sparsity of interactions depends on the value of $p$. Some special tasks heavily rely on the global information of all input variables, e.g., the task of judging whether the number of 1's in a binary sequence is odd or even. Then, in these tasks, the value of $p$ may be large. We have conducted **a new experiment** to train an MLP to judge whether the number of 1's in a binary sequence (e.g., the sequence [0,1,1,1,0,0,1,0,1,1]) is odd or even. Specifically, each binary sequence contains 10 digits. The MLP has 3 layers, and each layer contains 100 neurons. We find that the average value of $p$ was around 9.9 to 19.7, which was relatively large. We have also revised the paper to discuss this case in Scenario 5 of Section 3.3 and include this new experiment in Appendix E.3.
>
> Fortunately, in most classification tasks as shown in Figure 5(b), the model usually shows a certain level of robustness to the masking of input samples. Therefore, the value of $p$ will not be very large in most classification tasks, thus ensuring the sparsity of interactions encoded by the model.
>
> ---
>
> $\textcolor{blue}{\textsf{Q3: Ask about practicality of validating three conditions without empirical testing.}}$
>
> > "While intuitively, the proposed three criteria all make sense, however, for a general DNN model, how to validate these three conditions, besides empirically testing after training, is somehow unclear."
>
> A: Thank you. This study does not propose a method to sufficiently validate the sparsity of interactions in each specific DNN without empirically testing the DNN on masked samples. Instead, the main contribution of this study is that we have successfully proven the reason why the inference logic of most DNNs can all be faithfully explained as a small number of interactions. That is, we prove the sufficient conditions for the sparsity of interactions, and we find that these conditions are common in real applications.
>
> **Nevertheless, we have followed your suggestion to newly propose a method to efficiently examine the three conditions on each specific input sample, as follows.** In this way, we can prove the sparsity of interactions without a need of actually computing the interactions. We can simply sample a set of subsets (not all subsets) $\\{S_1,S_2,\cdots,S_t\\}$ for each order $m$ s.t. $\vert S_i\vert=m$, to approximate $\bar{u}^{(m)}$ and the value of $p$, which significantly reduces the computational cost. Mathematically, we have $\bar{u}^{(m)}\approx\frac{1}{t}\sum_{i=1}^t u(S_i)$, where $|S_i|=m$ for $1\le i \le t$. Then, we can simply use the above approximated $\bar{u}^{(m)}$ to compute the value $p$. In this way, the computational cost of empirically testing the monotonic increase of $\bar{u}^{(m)}$ and a rough estimation of the value of $p$ is $\mathcal{O}(nt)$, which is much less than $\mathcal{O}(2^n)$.
>
> We have also added Appendix I to discuss this technique.

---

> ### Author Response · Authors · 2023-11-17
> **Response to Reviewer 8DUx (Part 3)**
>
> $\textcolor{blue}{\textsf{Q4: Ask about the implication/benefit of our theoretical proof.}}$
>
> > "It would be nice if the authors could comment on implications of Thm 3, i.e. how would knowing many DNNs are able to learn a few key concepts benefit us? "
>
> A: In fact, the focus of our study, i.e., whether the inference logic of a DNN can be faithfully represented as symbolic primitives, is one of the fundamental problems in the field of explainable AI. Many previous studies have attempted to explain the feature representation of DNNs by using symbolic concepts with clear meanings. For example, Bau et al. [cite 2] tried to visualize the top-activated receptive fields corresponding to each convolutional filter to represent the concept encoded by the filter. Kim et al. [cite 3] assumed that there was a certain feature direction in the intermediate layer corresponding to each manually defined concept. These methods all potentially assumed that DNNs indeed encoded a small number of symbolic concepts rather than a massive number of fuzzy features. **Therefore, if the emergence of sparse interactions is disproved, then many previous methods will lose their theoretical support for the faithfulness of explaining DNN representations into semantic concepts**, although many studies have not formally discussed the faithfulness of the explanation. To this end, although it is hard to say our paper is the ultimate proof of the emergence of symbolic primitives in a DNN, our study proves that under three common conditions, a DNN encodes sparse symbolic interaction primitives. This can partially serve as somewhat support for previous attempts of semantic explanation of DNNs.
>
> In addition, some people have used interactions to redefine the generalization power of a DNN [cite 4]. The sparsity of interactions may further boost the trustworthiness of the generalization power explained by interactions in the studies.
>
> [cite 2] David Bau, Bolei Zhou, Aditya Khosla, Aude Oliva, and Antonio Torralba. Network Dissection: Quantifying Interpretability of Deep Visual Representations. CVPR, 2017.
>
> [cite 3] Been Kim, Martin Wattenberg, Justin Gilmer, Carrie Cai, James Wexler, Fernanda Viegas, and Rory Sayres. Interpretability Beyond Feature Attribution: Quantitative Testing with Concept Activation Vectors (TCAV). ICML, 2018.
>
> [cite 4] Huilin Zhou, Hao Zhang, Huiqi Deng, Dongrui Liu, When Shen, Shih-Han Chan, and Quanshi Zhang. Concept-Level Explanation for the Generalization of a DNN. arXiv preprint arXiv:2302.13091, 2023.

---

> ### Author Response · Authors · 2023-11-17
> **Response to Reviewer 8DUx (Part 4)**
>
> $\textcolor{blue}{\textsf{Q5: Ask about more experimental details.}}$
>
> > Q5.1: "Did the authors train the 5 mentioned models or existing trained model were used? "
>
> A: We used off-the-shelf trained models for all LLMs (including the OPT-1.3B model, the LLaMA-7B model, and the Aquila-7B model), which were all available on Huggingface. We used other models (including MLP, ResMLP, LeNet, AlexNet, VGG, PointNet, PointNet++) provided by [cite 5]. Nevertheless, our proof does not depend on the detailed setting of network training. Besides, we have revised the paper to clarify this detail in the first paragraph of Appendix C.1 and the first paragraph of Appendix D.1.
>
> > Q5.2: "Size of training, testing data. "
>
> A: **Size of training data:** We used off-the-shelf trained models for all experiments. For LLMs (including the OPT-1.3B model, the LLaMA-7B model, and BAAI Aquila-7B model), we used models provided by Huggingface. The training data for these models are specified by the corresponding web pages on Huggingface. Models on tabular datasets, the ShapeNet dataset, the MNIST-3 dataset, and the CelebA-eyeglasses dataset were provided by [cite 5]. All training samples in these datasets are used for training these models. Please see the second paragraph of Appendix C.1 for more details about the MNIST-3 dataset and the CelebA-eyeglasses dataset.
>
> **Size of testing data:** The heavy computational cost did not allow us to test on all the huge number of samples in datasets. Thus, similar to previous studies [cite 5], we used a subset of samples to compute interactions. For all models except for LLMs, we followed the setting of testing samples in [cite 5], in which 50 samples from the CelebA-eyeglasses dataset, 100 samples from the ShapeNet dataset and the MNIST-3 dataset, and 500 samples from each of the three tabular datasets have been selected to compute interactions. For LLMs, we tested on the first 1000 sentences in the SQuAD dataset. Please see the second paragraph of Appendix D.1 for details on these sentences.
>
> > Q5.3: "For each of the input, how was the subsets selected? For instance, for a image from MNIST, following the paper notation $X=(x_1,...,x_n)$, is each $x_i$ just a pixel? "
>
> A: For images in the MNIST-3 dataset, we used the semantic parts that had been annotated by [cite 5]. The annotation of different parts was based on some keypoints in each image. We have revised the paper to show annotated parts in some MNIST images. Please see Appendix C.2 and Figure 6 in the updated PDF for details.
>
>
>
> Furthermore, we have revised the paper to include all these details, in addition to the original experimental settings in Appendix C.1, Appendix C.2 and Appendix D.1.
>
> [cite 5] Mingjie Li and Quanshi Zhang. Does a Neural Network Really Encode Symbolic Concept? ICML, 2023.
>
> ---
>
> $\textcolor{blue}{\textsf{Q6: Ask about the value of $\lambda^{(k)}$ in Theorem 2 and the $p$'s bound in Theorem 3.}}$
>
> > "Based on eq (7), valid number of k-th order interactions is bound by roughly a constant times $\lambda^{(k)}*n^{p+\delta}$. On one hand, $\lambda^{(k)}$can be large. On the other hand, the authors claims $n^{p+\delta}$ is much less than $\binom{n}{k}$. Yet bound on $p$ is not required, in case where say $n-k<p$, not sure how the claim holds."
>
> A: Thank you. First, we have stated in Theorem 2 that $|\lambda^{(k)}|\le 1$, which provides an upper bound for $\lambda^{(k)}$. Please see Equation (48) in the proof for Theorem 2 in Appendix B.3 for how we obtain $|\lambda^{(k)}|\le 1$.
>
> Second, as mentioned in the answer to Q2, we have conducted experiments on Large Language Models (LLMs) to measure the true value of $p$ in real applications. Figure 5(b) shows that the value of $p$ across different samples was around 0.9 to 1.5. In this way, the claim that $n^{p+\delta}$ is much less than $\binom{n}{k}$ holds. Please see above answer to Q2 for more discussions on the value of $p$.
>
> In addition, we have revised the paper to make the claim $n^{p+\delta}$ is much less than $\binom{n}{k}$ clearer. Please see Figure 5(b) and the paragraph after Equation (7) in the updated PDF for details.
>
>
> **If our understanding of your questions is incorrect, please give us feedback as soon as possible. We will try our best to answer any further questions.**

---

> ### Comment · Reviewer_8DUx · 2023-11-21
> **Thank you for the clarification.**
>
> Thanks for your detailed responses , they mostly addressed my concern (a few points left are detailed below), and i have raised my score accordingly.
>
> I agree with the authors's assessment on Theorem 3 in their revision (new paragraph on page 8) that Theorem 3 shows that it is common for a DNN to encode sparse interactions. To make a strong argument, some bound on p needs to be derived.

---

> ### Comment · Reviewer_8DUx · 2023-11-21
> **Concerns on value of p**
>
> Thank you for the detail responses.
> There are a few points i am still confused on.
>
> - if a model satisfies assumption 2, then p must greater than 1, correct? then as the authors mentioned in the response, in some experiment p = 0.9. Is this because for some input, the assumption 2 does not hold or sampling error?
>
> - the new method the authors mentioned in the last paragraph of the response part 2, such tests are done after training, correct? or there are other ways to compute $u(S_i)$?

---

> ### Author Response · Authors · 2023-11-21
> **Response to further questions from Reviewer 8DUx**
>
> **Thank you very much for your feedback. We would like to carefully discuss and answer all your further concerns. If you are not satisfied with our answer, please give us feedback as soon as possible, so that we can get back to you in time.**
>
>
>
> $\textcolor{blue}{\textsf{Q7: Concerns on the value of $p$.}}$
>
> >If a model satisfies assumption 2, then p must be greater than 1, correct? Then as the authors mentioned in the response, in some experiment p = 0.9. Is this because for some input, the assumption 2 does not hold or sampling error?
>
> A: Thank you for your comments. First, we would like to clarify that if a model satisfies Assumption 2 (the monotonicity assumption), then $p$ is supposed to be greater than 0. It is not necessary to let $p$ be greater than 1. Therefore, the phenomenon that $p=0.9$ in some experiments does not contradict Assumption 2.
>
> In addition, we have revised the previous inaccurate presentation in Assumption 3 to clarify that $p>0$. Please see Assumption 3 in the revised paper for details.
>
> ---
>
> $\textcolor{blue}{\textsf{Q8: Ask about whether the assumptions can be examined before training the model.}}$
>
> > The new method the authors mentioned in the last paragraph of the response part 2, such tests are done after training, correct? Or there are other ways to compute $u(S_i)$?
>
> A: A good question. Theoretically, there **does not exist** any method to examine the sparsity of interactions encoded by the model **before** the training of the model. It is because the sparsity of interactions is not an inborn property of a model, but is determined by the type of the task and the optimization quality of the model. In most common tasks, a well-trained model is supposed to show certain robustness to the masking of input variables. In this way, the model will encode sparse interactions. In addition, we have discussed five specific types of tasks/scenarios that make the DNN encode relatively dense interactions in Section 3.3. This also shows that we cannot determine the sparsity of interactions before the model training.
>
> Therefore, the only way to examine the sparsity of interactions is to check whether the **trained** model follows our requirements, and we cannot examine the sparsity of the interactions before the model is trained. Nevertheless, we have proposed a new method to efficiently examine the satisfaction of the three conditions in our previous answer to Q3.
>
> ---
>
> $\textcolor{blue}{\textsf{Q9: Concerns on the monotonicity assumption.}}$
>
> >The authors mentioned in the last paragraph on page 6, quote"Note that this monotonicity assumption corresponds to a relatively ideal case. If many variables in the input sample represent the background or are not directly related to the target class, or if the model is not well-trained, then reducing the number of masked variables in the input sample may not necessarily increase the model’s classification confidence....."
> >My concern is along the same line. Since we can't really check this monotonicity prior training. Assumption 2 reads somewhat backward to me: if a model is well-trained, we sort of already know the model has learned something. Then the value of checking proposed assumptions is somewhat less important, as the authors do not have a bound on the number of concepts learned (no bound of p is proved).
>
> A: Thank you very much. The previous discussion in the last paragraph on Page 6 in the original paper is actually inaccurate, and we have revised the presentation to clarify the problem. Please see the paragraph "Justification of the monotonicity assumption" below Assumption 2 in the revised paper. In fact, even if there are input variables that represent the background or are unrelated to the classification, the monotonicity assumption still holds in most cases. Please see the answer to Q1 for more details. We have also added Appendix H in the revised paper to show an example with background input variables, in order to better explain your concern.
>
> In addition, the monotonicity of the average network output is determined by the task (the training objective) and the optimization quality of the model. There does not exist a way to examine this assumption before training. Please also see the answer to the above Q8 for more discussions.
>
>
>
> **Thank you again for your comments. If you have further questions, we would like to get back to you as soon as possible.**

---

> ### Comment · Reviewer_8DUx · 2023-11-22
>
> Dear authors,
>
> Thank you for clarify that the assumptions can only be checked after training, and general bound on p is yet to explore.
> Though i still have concerns on the actually impact,
> i would like to acknowledge the current theoretically contribution,
> the score has been raised accordingly.

---

### Official Review · Reviewer_WKdR · 2023-10-31

**Soundness:** 3 good
**Presentation:** 3 good
**Contribution:** 3 good
**Rating:** 8
**Confidence:** 2

**Summary:**

The main contribution of this paper is to provide another line of proof that AI models (at least those which are well trained) have (sparse) symbolic interactions emerge. In the end, the paper provides a proof that symbolic interactions can appear under a set of conditions on the AI model and its interaction with the data. Most importantly is using the Harsanyi dividend as the definition of interactions in an AI model. This along with several restrictions on the set of interactions with meaningful effect, how revealing more interactions ensures more confidence from the model, and bounding how poor the model performs on masked examples and how well it performs on unmasked examples.

**Strengths:**

This paper does a really nice job justifying their assumptions and explaining in detail where they come from. This is especially apparent in section 3.2 where each assumption is restated in less formal detail, and often experiments are performed to justify these in large language models.

**Weaknesses:**

I've increased my score accordingly from the conversation w/ the authors.


-----before edits---

This review is from an outsiders perspective, as I have very little experience in explainable AI. Outside of some experience in causality, I am primarily a reinforcement learning researcher. These weaknesses are a set of questions, which I believe limit the scope of work as presented. Overall, I think the paper is well written and provides many of the explanations and details I needed to understand and gain a reasonable intuition on the primary contribution.

I can’t speak well enough on the novelty and significance of this work, but I believe if others are satisfied the paper is sufficiently novel I will happily agree.

1. While using language is a decent entry to test the main properties of large models, this work (specifically how interactions are encoded) seems like it might be limited when working with models for image data. Specifically, how might one actually build a set of interactions (S) such that we achieve what is illustrated in figure 1?
2. While the paper says these are properties of a well trained AI model, don’t these assumptions really speak more to the data you are using to train and test the AI model? Assumption 2 does speak towards the output of the model, but isn’t this necessarily implying the input data can be well partitioned into the discrete set of interactions?
3. Assumption 3 is the weakest in terms of justification. While it applies to our current models, I’m not sure the upperbound on inference confidence is a property generally exhibited by “AI models”. This wouldn’t be an issue if this was stated as a part of the discussion of assumption 3, but I don’t buy the upperbound portion being undesirable.


### Edits
- I think you should mention the relationship to shapley values a bit more prominently in your main text. You have the relations expanded in the appendix, but as a sentence of a footnote should include a point to this in your main text (I was distracted thinking how much this reminded me of Shapley values, but was not familiar with Harsanyi interactions).
- Figures 1 and 2 should be moved closer to where they are referenced in the paper.
- section 3.1: “sparse symbolic (spase) interactions” I think the first sparse should be removed?
- 3.2: “Assumption 1-\alpha” you can use \boldsymbol{\alpha} to make the symbol bolded in the assumption header.

**Questions:**

1. How limiting is the focus on Harsanyi interactions in your opinion? While I believe this focus is ok for input such as an image or textual data (which are by nature sub-dividable), can this be applied to real valued inputs? This is of particular interest in many applications such as time-series forecasting, reinforcement learning and control, etc…

---

> ### Author Response · Authors · 2023-11-20
> **Response to Reviewer WKdR (Part 1)**
>
> Thank you very much. We have found that many comments have deep insights. We are glad to answer all your questions.
>
> **If you are not satisfied with our answers or have more questions, please let us know as soon as possible, so that we can try our best to answer any further questions before the deadline.**
>
> $\textcolor{blue}{\textsf{Q1: Ask about how to build a set of interactions.}}$
>
> > "this work (specifically how interactions are encoded) seems like it might be limited when working with models for image data. Specifically, how might one actually build a set of interactions (S) such that we achieve what is illustrated in figure 1? "
>
> A: A good question. We would like to answer this question from four perspectives. **First, before detailed answers on the building of interactions, we need to clarify that the objective of this paper, i.e., the proof of the emergence of sparse interactions, is agnostic to the selection of input variables for interactions.** For example, we can view words or tokens in a sentence as input variables, and we can use image patches of different sizes as input variables.  To better illustrate this, we have conducted **a new experiment** on the MNIST dataset to compare the sparsity of interactions when we set different sizes of image patches as input variables. Specifically, in our experiments on the MNIST dataset in Figure 3, we used the image patches annotated by Li & Zhang (2023) as the input variables. Original image patches annotated by Li & Zhang (2023) were of the size of $3\times 3$ pixels. Then, we enlarged the size of image patches to $5\times 5$ pixels, and $7\times 7$ pixels. Figure 13 in Appendix E.4 in the revised paper compares the normalized interaction strength in descending order under different settings of the size of image patches. It indicates that under image patches of different sizes, we still extracted sparse interactions. We have clarified this and added the above experiments in Appendix E.4 in the revised paper.
>
> **Second, in real applications, let us introduce how to set input variables.** Please see Appendix C.2 for the setting of input variables in image data, and see the second paragraph of Appendix D.1 for the setting of input variables in language data. Because the computational cost is $\mathcal{O}(2^n)$ given an input sample with $n$ input variables, we used input variables annotated by previous works (Li & Zhang, 2023; Shen et al., 2023) to speed up the computations.
>
> **Third, considering your question, we have proposed a new method to identify important input variables.** Please see the third paragraph in Appendix D.1 for details. Because the computational cost of all interactions is $\mathcal{O}(2^n)$, the most straightforward solution is to select a small set of important input variables for the task, so as to only compute interactions between the selected variables and speed up the computation. Specifically, we can select input variables with significant Shapley values. Theorem 4 shows that $\phi(i)=\sum_{S\subseteq N\setminus \\{i\\}}\frac{1}{|S|+1} I(S\cup \\{i\\})$. It means that the Shapley value $\phi(i)$ of the input variable $i$ can be explained as the result of uniformly assigning each Harsanyi interaction $I(S\cup \\{i\\})$ to each involved variable. Therefore, the Shapley value can serve as a reasonable metric to measure the saliency of input variables. We can select a set of input variables with large absolute values of Shapley values, $\vert\phi(i)\vert$. Fortunately, in most real applications, only a few input variables are directly related to the network output and have considerable Shapley values. Most other input variables have negligible Shapley values. For example, in object detection, only patches around the target object have large attention/importance. Thus, we usually do not need to use a large number of input variables.
>
> **Finally, let us introduce how to build up interactions when input variables are given.** Please see Equation (1) in the main paper for how to compute the interaction effects of each subset $S$. Figure 3 shows the strength of all interactions in descending order. In this way, we can set a threshold $\tau$ to select salient interactions (i.e., interactions with $|I(S)|>\tau$). Please see the paragraph right above Section 3.2 for the detailed setting of $\tau$.
>
> (Li & Zhang, 2023) Mingjie Li and Quanshi Zhang. Does a Neural Network Really Encode Symbolic Concept? ICML, 2023.
>
> (Shen et al., 2023) Wen Shen, Lei Cheng, Yuxiao Yang, Mingjie Li, and Quanshi Zhang. Can the Inference Logic of Large Language Models be Disentangled into Symbolic Concepts? arXiv preprint arXiv:2304.01083, 2023.

---

> ### Author Response · Authors · 2023-11-20
> **Response to Reviewer WKdR (Part 2)**
>
> $\textcolor{blue}{\textsf{Q2: Do our assumptions focus on the data or the model?}}$
>
> > "While the paper says these are properties of a well-trained AI model, don’t these assumptions really speak more to the data you are using to train and test the AI model?  "
>
> A: This is a quite insightful comment. This question actually asks about the essential reason for the emergence of sparse interactions in a DNN. **First, the three assumptions directly constrain the model outputs, rather than other factors (such as the architecture of a DNN).** They requires the model not to encode high-order derivatives (or high-order interactions), and to conduct smooth inference on masked samples. If these conditions are satisfied, the network will encode sparse interactions.
>
> **Second, the task itself (or, more precisely, the training data) is the real reason why a model is trained to satisfy the above three assumptions.** For most classification tasks, the need for the classification on missing data (with masked/occluded input variables) makes a well-trained DNN conduct smooth inference on different masked samples. Because it is hard to ensure that every training sample perfectly contains all important variables without being occluded, a well-trained DNN is usually supposed not to encode very high-order interactions (between almost all input variables) and to conduct smooth inferences on masked/occluded samples.  We have briefly discussed this in the last paragraph of the introduction section.
>
> **Nevertheless, we have also shown several specific tasks that cannot achieve sparse interactions.** Please see Section 3.3 for details.
>
> ---
>
> $\textcolor{blue}{\textsf{Q3: Ask about the justification of Assumption 3.}}$
>
> > " Assumption 3 is the weakest in terms of justification. While it applies to our current models, I’m not sure the upperbound on inference confidence is a property generally exhibited by 'AI models'. "
>
> A: Thank you very much for your suggestions. We have revised the discussion below Theorem 3, and we have emphasized that we have not strictly proven the upper bound for the value of $p$.
> The value of $p$ in Assumption 3 depends on the task. **In fact, for DNNs trained for most tasks, we usually obtain relatively small $p$ values.** We have conducted some experiments on LLMs in Figure 5(b), which shows that the value of $p$ across different samples was around 0.9 to 1.5. According to both Theorem 3 and experimental verification in Table 2, interactions encoded by these LLMs were sparse. In addition, **Table 2 shows that the proven upper bound just represents the worst case in theory**, and the real number of valid interactions was usually much less than the theoretical upper bound. Thus, the $p$ value is usually not a threat for the sparsity of interactions.
>
> More crucially, we have conducted **a new experiment** in Appendix E.3 to **show a special task** that heavily relies on the global information of all input variables, **which leads to a large $p$ value**. Specifically,  we trained an MLP for the task of judging whether the number of 1's in a binary sequence (e.g., the sequence [0,1,1,1,0,0,1,0,1,1]) is odd or even. Specifically, each binary sequence contains 10 digits. The MLP has 3 layers, and each layer contains 100 neurons. We find that the average value of $p$ was around 9.9 to 19.7, which was relatively large.
>
> Nevertheless, the proposed three assumptions are just sufficient conditions for the upper bound of the number of salient interactions. **It means that the DNN may still encode sparse interaction even when these assumptions are not satisfied.** Specifically, we have shown in the third row of Table 1 that we could obtain sparse interactions even on samples that did not satisfy the monotonicity assumption.
>
> ---
>
> $\textcolor{blue}{\textsf{Q4: Ask to mention the relationship between the Harsanyi interaction and the Shapley value in the main text.}}$
>
> > "I think you should mention the relationship to shapley values a bit more prominently in your main text. You have the relations expanded in the appendix, but as a sentence of a footnote should include a point to this in your main text. "
>
> A: Thank you. We have followed your suggestion to add a footnote (i.e., the footnote 2) to briefly introduce the relationship between the Harsanyi interaction and the Shapley value in the main text. Please see the paragraph below Theorem 1 for details. The detailed relationship is introduced in Theorem 4 in Appendix A, and is proven in Appendix B.5. Specifically, the theorem shows that $\phi(i)=\sum_{S\subseteq N\setminus \\{i\\}}\frac{1}{|S|+1} I(S\cup \\{i\\})$. It means that the Shapley value $\phi(i)$ of the input variable $i$ can be explained as the result of uniformly assigning each Harsanyi interaction $I(S\cup \\{i\\})$ to each involved variable.

---

> ### Author Response · Authors · 2023-11-20
> **Response to Reviewer WKdR (Part 3)**
>
> $\textcolor{blue}{\textsf{Q5: Ask to reorganize the position of figures.}}$
>
> > "Figures 1 and 2 should be moved closer to where they are referenced in the paper.  "
>
> A:  Thank you. We have followed your suggestion to move Figure 2 closer to where it is referenced.
>
> ---
>
> $\textcolor{blue}{\textsf{Q6 and Q7: Typos and formatting issues.}}$
>
> > "Section 3.1: “sparse symbolic (spase) interactions” I think the first sparse should be removed?  "
>
> >"3.2: “Assumption 1-\alpha” you can use \boldsymbol{\alpha} to make the symbol bolded in the assumption header. "
>
> A: Thank you. We have followed your suggestion to correct the above typos.
>
> ---
>
> $\textcolor{blue}{\textsf{Q8: Ask whether the Harsanyi interaction can be applied to real-valued inputs.}}$
>
> > "While I believe this focus is ok for input such as an image or textual data (which are by nature sub-dividable), can this be applied to real valued inputs?  This is of particular interest in many applications such as time-series forecasting, reinforcement learning and control, etc…  "
>
> A: A good question. We have followed your suggestion to conduct **a new experiment** on an RNN (i.e., the LSTM) to explore the possibility of applying the Harsanyi interaction to a wider range of tasks. Specifically, we trained LSTMs with 2 layers on the SST-2 dataset (for sentiment classification) and the CoLA dataset (for linguistic acceptability classification). The LSTM can be considered to take natural language as sequential data. Although these tasks are not equivalent to other prediction tasks on sequential data, they provide potential insights into how we can use the Harsanyi interactions to explain DNNs on sequential data. Figure 14 in Appendix E.5 shows that the network encodes relatively sparse interactions. Please see Appendix E.5 in the revised paper for details.
>
> In fact, do you want to say "real-time input data," instead of "real-valued inputs"? If you are not satisfied with our feedback or we misunderstand your question, please let us know as soon as possible, so that we can get back to you early.

---

> > ### Comment · Reviewer_WKdR · 2023-11-22
> > **Response**
> >
> > Thank you for your response! I am happy with your additions and edits.
> >
> > Again, I am not an expert in this area, but am happy with the paper as it currently stands.

---

### Author Response · Authors · 2023-11-20
**Response to all Reviewers**

We would like to thank all reviewers for their constructive comments and questions. This paper has received diverse ratings of **8** (acceptance), **6** (borderline acceptance), **5** (borderline reject), and **3** (reject). We have carefully considered all your comments and answered all the questions, and have revised the paper to clarify all your concerns. In addition, we have followed your suggestions to conduct new experiments.

**Please let us know if you are not satisfied with our answers or have further questions, so that we can get back to you as soon as possible.**

---

### Author Response · Authors · 2023-11-22
**Thanks to all Reviewers**

Thank you very much for your timely feedback. We are glad to see that most concerns are addressed during the discussion period. If you have further questions, please let us know, and we will try to get back to you as soon as possible.

---

### Meta-Review · Area_Chair_jAks · 2023-12-11

**Metareview:**

This paper examines whether well-trained DNNs can be shown to possess symbolic concepts. The authors do this by first defining a set of 3 sufficient conditions based on sparse interactions between the input variables, along with proofs and derivations, and devising a range of experiments to test these assumptions on various neural networks.

The main reviewer comments concerned questioning the validity of the 3 assumptions as well as clarification questions. During the rebuttal period, reviewers and authors engaged in lengthy discussions, and in response the authors added several new experiments and detailed, comprehensive answers. As a result, reviewers seemed mostly satisfied, with 3 out of 4 reviewers raising their score, and there was a general consensus that this is an impactful paper that should be accepted.

As this seems to be a well-written paper that has sparked interest in all the reviewers, who agree it makes a solid contribution, I’m happy to recommend acceptance.

**Justification For Why Not Higher Score:**

I'm not sure the strength of the contributions really propels this up to a spotlight, as it's not entirely clear what the wider implications or applications of this work would be. It would have been nice to see it applied concretely within an explainable AI setting.

**Justification For Why Not Lower Score:**

There was a consensus among reviewers to accept.

---

### Decision · Program_Chairs · 2024-01-16

Accept (poster)